# Ligand-induced transmembrane conformational coupling in monomeric EGFR

Shwetha Srinivasan[1,7], Raju Regmi [1,5,7], Xingcheng Lin [1], Courtney A. Dreyer[2], Xuyan Chen[1], Steven D. Quinn[1,6], Wei He [3], Matthew A. Coleman[3,4], Kermit L. Carraway III[2], Bin Zhang [1✉] & Gabriela S. Schlau-Cohen [1✉]

Single pass cell surface receptors regulate cellular processes by transmitting ligand-encoded signals across the plasma membrane via changes to their extracellular and intracellular conformations. This transmembrane signaling is generally initiated by ligand binding to the receptors in their monomeric form. While subsequent receptor-receptor interactions are established as key aspects of transmembrane signaling, the contribution of monomeric receptors has been challenging to isolate due to the complexity and ligand-dependence of these interactions. By combining membrane nanodiscs produced with cell-free expression, single-molecule Förster Resonance Energy Transfer measurements, and molecular dynamics simulations, we report that ligand binding induces intracellular conformational changes within monomeric, full-length epidermal growth factor receptor (EGFR). Our observations establish the existence of extracellular/intracellular conformational coupling within a single receptor molecule. We implicate a series of electrostatic interactions in the conformational coupling and find the coupling is inhibited by targeted therapeutics and mutations that also inhibit phosphorylation in cells. Collectively, these results introduce a facile mechanism to link the extracellular and intracellular regions through the single transmembrane helix of monomeric EGFR, and raise the possibility that intramolecular transmembrane conformational changes upon ligand binding are common to single-pass membrane proteins.

[1] Department of Chemistry, Massachusetts Institute of Technology, 77 Massachusetts Avenue, Cambridge, MA 02139, USA. [2] Biochemistry and Molecular Medicine, University of California Davis School of Medicine, Sacramento, CA 95817, USA. [3] Lawrence Livermore National Laboratory, Livermore, CA 94550, USA. [4] Radiation Oncology, University of California Davis School of Medicine, Sacramento, CA 95817, USA. [5] Present address: Institut Curie, CNRS, Laboratoire Physico Chimie Curie, Paris, France. [6] Present address: Department of Physics, University of York, York, UK. [7] These authors contributed equally: Shwetha Srinivasan, Raju Regmi. ✉email: binz@mit.edu; gssc@mit.edu

Receptor tyrosine kinases, surface receptors present in all cell types across the animal kingdom, regulate major cellular functions, including cell division and survival[1–3]. The regulatory signals are primarily initiated by extracellular ligand binding to monomeric receptors, which causes intracellular autophosphorylation and subsequent recruitment of adapter proteins to the phosphorylated residues[1]. Epidermal growth factor receptor (EGFR), a prototypical receptor tyrosine kinase, has been extensively investigated as its aberrant expression leads to diseases such as cancer and diabetes[4,5]. Binding of its most studied ligand, the epidermal growth factor (EGF), induces a conformational expansion of the extracellular region, enabling dimerization of EGFR[6,7]. This expansion as well as other ligand-induced changes have been well characterized for the extracellular region[8,9]. The corresponding changes to the intracellular region, however, have only been accessible for oligomers due to the limited window between ligand binding and dimerization[10]. Analysis of fragmented domains has emerged as an alternative strategy that can isolate the conformations associated with signaling states[11–16], yet these domains cannot be used to visualize how extracellular stimuli are propagated across the plasma membrane such as through extracellular/intracellular conformational coupling[17]. In 47% of membrane proteins, including EGFR, a single transmembrane helix spans the plasma membrane[18]. Although different conformations of this helix alone have been observed[19–21], how, or even whether, the single helix can support extracellular/intracellular conformational coupling to mediate a signaling cascade has been largely unexplored.

Prior to ligand binding, 95% of EGFR is found in its monomeric form in cells[22]. While EGF-induced dimers have been long established as an active form of the receptors, emerging evidence suggests that other oligomerization states of EGFR also play a role in phosphorylation and signaling[6,23,24]. Both homo- and heterodimerization between members of the EGFR family has been observed[25,26]. The nature of the dimer can enhance ligand affinity or protein binding, providing an alternative mechanism to control signaling levels[27–30]. Depending on the lipid composition of the plasma membrane, ligand-induced formation of multimers induces stronger and more complete phosphorylation of the tyrosines and a wider dynamic range of EGFR responsiveness[24,31–33]. Furthermore, early studies suggested EGFR signaling can occur even in the presence of an antibody that prevented dimer and multimer formation[23,34]. Consistently, no homodimerization was observed for the ligand epigen, yet it still induces signaling[35–37]. Despite these multiple lines of evidence, the behavior of monomeric EGFR prior to oligomerization and its contribution, if any, to the signaling pathway have not yet been determined. Here, we use a multidisciplinary approach involving mutagenesis, single-molecule Förster resonance energy transfer (smFRET), molecular dynamics (MD) simulations, and cellular phosphorylation studies to isolate and investigate extracellular/intracellular conformational coupling within monomeric EGFR and its impact on cell signaling.

## Results

**Labeled EGFR monomers in nanodiscs.** Cell-free expression was used to produce full-length EGFR monomers embedded in lipid bilayer nanodiscs and free of cellular interaction partners (Fig. 1a, Supplementary Figs. 1–3)[38]. A FRET donor dye (snap surface 594) was covalently attached to the C-terminus of the protein and an acceptor dye (Cy5) was introduced as a labeled lipid within the bilayer (Supplementary Fig. 4)[39]. The functional, folded conformation of the labeled receptors was implied with ATP-dependent phosphorylation for the intracellular region (Supplementary Fig. 2, Supplementary Tables 1, 2) and specificity of

ligand binding for the extracellular region (Fig. 2), consistent with previously published western blot and fluorescence-based phosphorylation assays with similar preparations[38,39]. Intact, full-length monomeric EGFR was further purified spectroscopically by immobilizing nanodiscs on a coverslip at dilute concentration and only selecting receptors with a single donor and acceptor for analysis (Fig. 1b, Supplementary Fig. 5).

**Intracellular conformational change.** The intracellular conformation was examined with smFRET by measuring the fluorescence lifetime of the donor. FRET, which depends on the distance between the donor and acceptor dyes, competes with emission, thereby shortening the lifetime in a distance-dependent manner[40]. The fluorescence lifetime of the donor was measured in the absence and presence of saturating (1 µM) EGF ligand. To first benchmark the behavior, monomeric wild-type (WT) EGFR was embedded in a nanodisc with a 1,2-Dimyristoyl-sn-glycero-3-phosphocholine (DMPC) bilayer, which is neutrally charged and thus lacks the complex electrostatic interactions of the in vivo plasma membrane. For this sample, a shorter (2×) fluorescence lifetime of the donor was observed in the presence of EGF compared to in its absence for most receptors, signifying a decrease in distance between the C-terminus and the membrane surface upon EGF binding (Fig. 1c).

We built donor lifetime histograms for WT EGFR with and without EGF ligand (Fig. 2a). The corresponding donor-acceptor distances were calculated from the lifetimes with reference time $t_D = 3.32$ ns (Supplementary Fig. 5c). In the absence of EGF, the distribution peaked at ~3 ns (12 nm), whereas in the presence of EGF, the distribution peaked at ~1.5 ns (~8 nm; Supplementary Figs. 6–8 and Supplementary Table 3). A structural model of active EGFR dimers suggests that the kinase domain (KD) of one monomer is lifted towards the membrane, similar to the change seen in the smFRET measurements of monomers[10].

Fluorescence correlation spectroscopy (FCS) was performed on diffusing donor-only constructs of the full-length, nanodisc-embedded EGFR to characterize the overall structural change of the receptor-embedded nanodisc (Fig. 3a; also see "Methods"). The diffusion coefficient, which is a function of the hydrodynamic radius, can be extracted from transit times through a known confocal volume[39]. Diffusion coefficients of $(1.27 \pm 0.08) \times 10^{-7}$ cm$^2$ s$^{-1}$ and $(0.98 \pm 0.03) \times 10^{-7}$ cm$^2$ s$^{-1}$ were found in the presence and absence of EGF, respectively, corresponding to a compaction of the hydrodynamic radius (~22%) upon EGF binding (Fig. 3a). Previous crystallographic studies of EGFR reported an expansion in the extracellular region upon EGF binding[41]. Therefore, the observed compaction likely originates in the intracellular region, consistent with the smFRET results shown in Fig. 2a.

We also investigated the effect of other EGFR ligands, beginning with Cetuximab, an EGFR antibody administered for metastatic colorectal cancer that inhibits ligand-induced phosphorylation and signaling[42]. Cetuximab binds at the same extracellular site as EGF but does not cause an extracellular expansion[22]. In the presence of 100 nM Cetuximab together with 1 µM EGF (Fig. 2a, third), the lifetime distributions peaked ~3 ns (12 nm), similar to the distribution in the absence of EGF (Supplementary Figs. 6–8). In FCS measurements, addition of the Cetuximab with EGF produced a diffusion coefficient of $(1.10 \pm 0.03) \times 10^{-7}$ cm$^2$ s$^{-1}$ (Fig. 3a, cyan), which approaches the value in the absence of EGF. The correlation between the extracellular expansion and intracellular compaction indicates the presence of extracellular/intracellular conformational coupling.

While the neutral bilayer provided the simplest environment for initial experiments, we then ascertained the effect of a

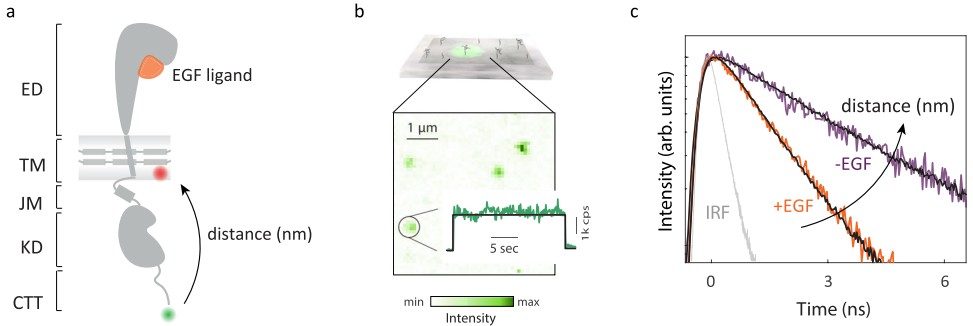

**Fig. 1 smFRET measures intracellular conformational states of full-length EGFR in a nanodisc. a** Full-length, monomeric EGFR (solid gray) embedded in a nanodisc. The nanodisc is a lipid bilayer (shaded gray) belted by an amphiphilic apolipoprotein (solid gray). EGFR consists of a 621-amino acid extracellular region (ED) that binds EGF (orange), a 24-amino acid transmembrane-spanning domain (TM), and an intracellular region, which is a 37-amino acid juxtamembrane domain (JM), a 273-amino acid kinase domain (KD) and a 231-amino acid disordered C-terminal tail (CTT) (Supplementary Fig. 1). Green and red spheres indicate the donor and acceptor dyes, respectively. **b** Top: Ni-NTA coated coverslip binds EGFR nanodiscs via a His-tag on the apolipoprotein. Bottom: fluorescence intensity from a confocal image for a representative region ($\lambda_{exc} = 550$ nm) where green spots are immobilized EGFR nanodiscs. Number of detected photons for each 100 ms interval generates a fluorescence intensity trace (green) with the average intensity indicated (black). **c** Histogram of the arrival times of detected photons generates the donor lifetime decay profile. Representative decay profiles of EGFR in the presence (orange) and absence (purple) of the EGF ligand in a neutral bilayer with fit curves (black). The instrument response function (IRF) is shown in gray.

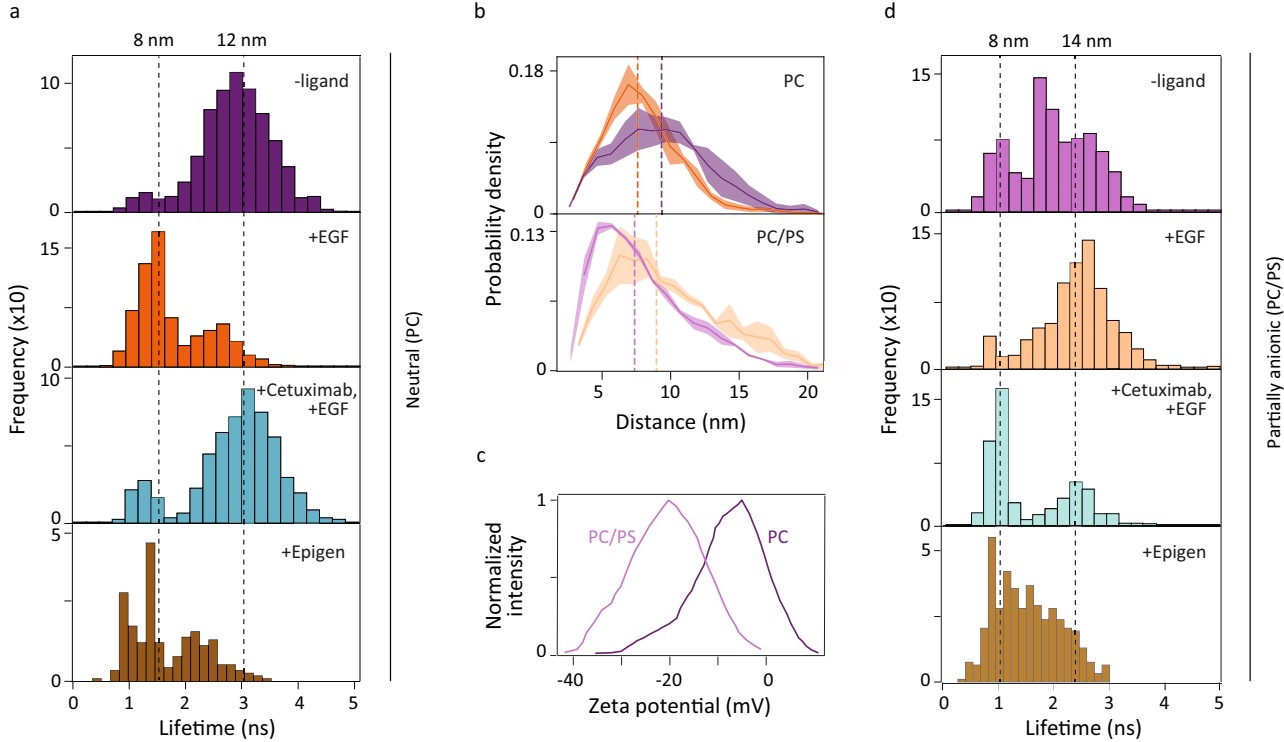

**Fig. 2 smFRET and MD simulations reveal extracellular/intracellular conformational coupling in EGFR.** smFRET measurements (**a**, **d**) were used to build histograms of the donor lifetime, which shortens as the donor-acceptor distance decreases. Dotted lines indicate the medians with corresponding distances on upper x-axis. **a** The donor fluorescence lifetime distribution in DMPC (PC) bilayer without ligand (top); with 1 μM EGF (second); with 1 μM EGF and 100 nM Cetuximab (third); with 1 μM epigen (bottom). **b** Molecular dynamics trajectories were used to find the probability density distribution of the vertical separation in a PC bilayer (top) or a POPC-POPS (PC/PS) bilayer (bottom) without EGF (purple) and with EGF (orange). Dotted lines indicate the median of the distributions (PC bilayer: 9.2 nm (-EGF) and 7.5 nm (+EGF); PC/PS bilayer: 7.4 nm (−EGF) and 9.0 nm (+EGF)). The shaded regions represent the standard deviation estimated with block averaging. **c** Zeta potential distributions for EGFR in PC (dark purple) and PC/PS (light purple) nanodiscs. **d** The donor fluorescence lifetime distribution in PC/PS bilayer without ligand (top); with 1 μM EGF (second); with 1 μM EGF and 100 nM Cetuximab (third); with 1 μM epigen (bottom). Source data are provided as a Source data file.

near-native lipid composition on the intracellular region by incorporating a partially anionic bilayer into the nanodiscs (70% 1-palmitoyl-2-oleoyl-sn-glycero-3-phosphocholine, POPC; 30% 1-palmitoyl-2-oleoyl-sn-glycero-3-phospho-L-serine, POPS; Fig. 2c, Supplementary Fig. 3), which replicates the plasma membrane anionic lipid content of mammalian cells[43]. The electrostatic interactions introduced by these anionic lipids replicates this important aspect of the cellular environment, which have been previously implicated in the regulation of EGFR signaling[44,45]. As shown in Fig. 2d, smFRET measurements were

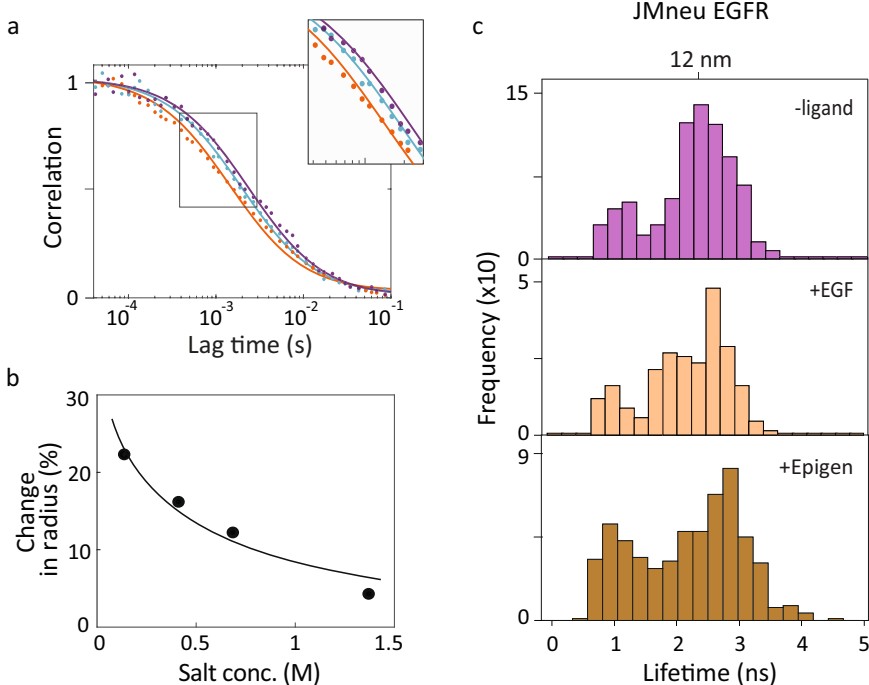

**Fig. 3 Charged residues implicated in extracellular/intracellular conformational coupling. a** FCS curves of WT EGFR (dots) were fit (solid lines) to extract a diffusion time of 1.5 ms in the presence of EGF (orange), 2.2 ms in the absence of EGF (purple), corresponding to a reduction of 22% in hydrodynamic radius, and 1.9 ms in the presence of Cetuximab together with EGF (cyan). **b** Percent change in hydrodynamic radius from FCS curves in the presence of EGF relative to in the absence of EGF (dots) as a function of salt concentration with Debye-Huckel fit curve (solid line). **c** smFRET donor fluorescence lifetime histograms for JMneu EGFR in POPC-POPS bilayer without ligand (top); with 1 μM EGF (second); with 1 μM epigen (bottom). All the three sets of histograms are statistically similar to each other (Supplementary Fig. 14). Source data are provided as a Source data file.

performed on the receptors in the partially anionic lipid bilayer for all four ligand conditions. In the absence of EGF, the distribution was broad and structured with a median at ~2 ns (10 nm), whereas in the presence of EGF, the distribution appeared closer to unimodal and peaked at ~2.50 ns (≥13 nm; Supplementary Fig. 9 and Supplementary Table 3). Although an EGF-induced intracellular conformational change was again observed, these distributions showed an expansion of the intracellular region, in contrast to the results with the neutral bilayer. In the presence of Cetuximab together with EGF, the distribution peaked at ~1.1 ns (8 nm), which indicated a more compact intracellular region, although one that was again closer to the conformation in the absence of EGF than in its presence. The additional compaction induced by Cetuximab may interfere with the access of substrates and/or signaling proteins to the intracellular binding sites. The donor lifetime distributions appear bimodal, consistent with a model in which a pre-existing equilibrium between at least two conformations shifts in the presence of the ligand[46,47].

To establish the generality of the extracellular/intracellular conformational coupling of EGFR, we investigated the effect of epigen, a ligand that belongs to the EGF family yet does not induce homodimerization. In the presence of 1 μM epigen, the lifetime distribution peaked at ~1.5 ns (8.5 nm), indicating an intracellular compaction was observed for both bilayers (Fig. 2a, d, bottom). While the specific conformational response is ligand-dependent[8,48], these distributions establish that the extracellular/intracellular conformational coupling observed through smFRET is not uniquely induced by EGF.

**Effect of electrostatic interactions.** Given the pronounced effect of anionic lipids on the nature of ligand-induced intracellular

conformational changes, we used mutagenesis to investigate the role of electrostatic interactions in extracellular/intracellular conformational coupling. First, we used all atom MD simulations with the CHARMM force field to implicate two regions in the intracellular conformational change (Supplementary Fig. 10), the juxtamembrane domain (JMD), which is the membrane adjacent region of the intracellular domain, and the C-terminal tail (CTT), which contains the tyrosine phosphorylation sites. Next, we exchanged charged residues with their neutral analogs for these regions. To examine the contribution of each domain individually, six of the positively-charged residues in the JMD were neutralized in one mutant (JMneu EGFR) and five of the negatively charged residues in the CTT were neutralized in a second (CTTneu EGFR; see "Methods", Supplementary Fig. 11).

We performed smFRET measurements on both mutants in neutral and partially anionic lipid bilayers and in the presence and absence of EGF and of epigen. We built donor lifetime histograms for the ten samples (Fig. 3c and Supplementary Figs. 12, 13). In contrast to the distributions for WT EGFR (Fig. 2a, d), the distributions for the mutant samples were statistically similar in the absence and presence of EGF and of epigen in both bilayers (Supplementary Figs. 14, 15), indicating that the ligand-induced intracellular conformational changes are suppressed upon mutation. For JMneu EGFR in both lipid bilayers and CTTneu EGFR in a neutral bilayer, the distributions peaked at ~12 nm, which is the same distance as for WT EGFR in a neutral bilayer in the absence of EGF; this likely reflects a similar intracellular conformation due to loss of ligand-dependent electrostatic interactions. For CTTneu EGFR in a partially anionic bilayer, the distributions peaked at ~1.2 ns (8 nm), where the decreased distance may be attributed to a loss of repulsion between the negative residues on the tail and anionic bilayer. In contrast to what we observe for WT EGFR, previous structural

studies did not show an intracellular conformational change upon EGF binding[49]. However, the EGFR construct employed in these experiments lacked the CTT, and thus may be more analogous to the measurements of CTTneu EGFR, where an EGF-induced conformational change is similarly not observed.

As additional characterization of the role of electrostatic interactions in the conformations of WT EGFR, we used FCS to measure the EGF-induced compaction of the hydrodynamic radius in a neutral bilayer as a function of salt concentration (Fig. 3b). We observed that the 22% compaction at physiological salt concentration (137 mM) reduced to 4% at high salt concentration (1.37 M). The data points as a function of salt concentration were fit using Debye-Huckel theory for electrostatic screening[50], which gave good agreement with the measured values. These results support a model in which extracellular/intracellular conformational coupling is driven by electrostatic interactions.

**Simulations capture measured distances.** To further investigate the mechanism behind the observed extracellular/intracellular conformational coupling, we performed explicit solvent MD simulations on full-length, monomeric EGFR. Simulations were carried out on ligand bound (+EGF) or ligand unbound (−EGF) conformations in neutral (DMPC) or partially anionic (70% POPC/30% POPS) lipid bilayers for a total of 400 μs using a calibrated Martini force field with improved accuracy in modeling the disordered CTT ("Methods", Supplementary Fig. 16)[51].

The average vertical separations between the center of mass of the membrane and the C-terminus (residue 1186) were used to compare the simulated structures with the smFRET results (Fig. 4a). In the neutral bilayer, the average vertical separation stabilized at 9.2 and 7.9 nm for the simulated conformations in the absence and presence of EGF, respectively (Fig. 2b, top). In contrast, in the partially anionic bilayer, the average vertical separation was longer in the presence of EGF (9.8 nm) than in its absence (8.1 nm; Fig. 2b, bottom). Therefore, comparison amongst the intracellular domains of the simulated structures showed that the simulations succeeded in reproducing the experimental trends (Fig. 2b, Supplementary Fig. 17).

**Transmembrane conformational coupling.** Collectively, the simulations indicate that the overall intracellular conformation can be characterized based on two parameters: the distance between the N-terminal portion of the CTT (NCTT) and the plasma membrane and the number of contacts between the C-terminal portion of the CTT (CCTT) and the KD (Supplementary Fig. 18), as illustrated in Fig. 4a. For the structures that corresponded to the smFRET measurements, both of these parameters were strongly correlated with the measured distances (Supplementary Figs. 18–20). Examination of the simulations, along with the experimental data and previously published results, points to a molecular mechanism for extracellular/intracellular conformational coupling, as illustrated in Fig. 4b, c.

With EGF bound, the extracellular domain extends above the membrane[45], imposing a vertical orientation to the transmembrane domain (TMD) (Supplementary Fig. 21). Without EGF, the hinge action of the extracellular domain causes it to lay flat on the membrane[45], imposing a tilted orientation to the TMD

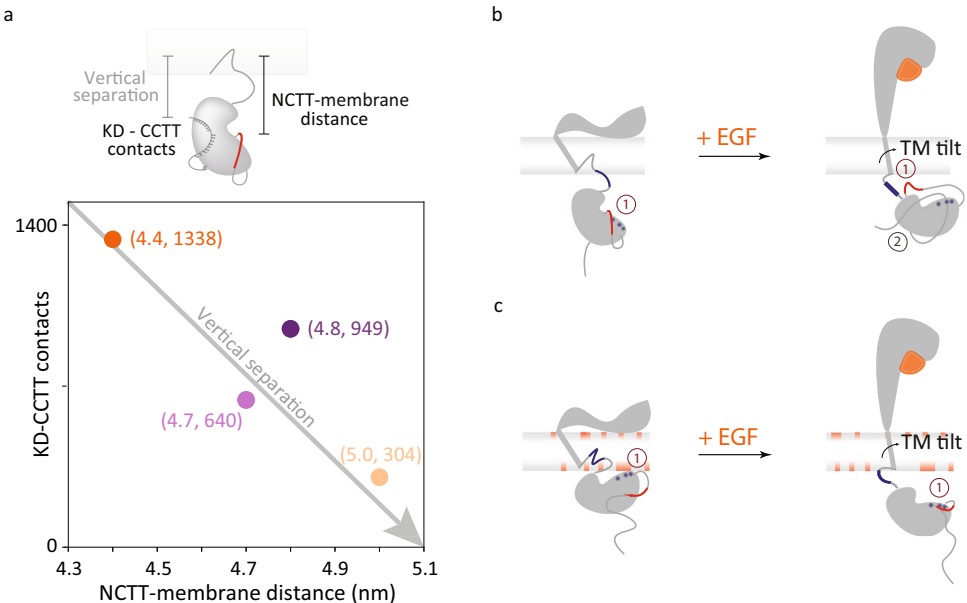

**Fig. 4 Intracellular conformational changes upon extracellular EGF binding. a** The intracellular conformation can be characterized by the distance between the NCTT and the membrane and the contacts between the CCTT and the KD. The positions of WT EGFR in both neutral (dark colors) and partially anionic (light colors) lipid bilayers in the absence (purple) and presence (orange) of EGF are plotted as a function of the average NCTT-membrane distance and KD-CCTT contacts obtained from simulations. The average vertical separation between the center of mass of the membrane and residue 1186 from simulations is used to compare with the smFRET measurement. **b, c** Schematic of proposed mechanism for EGFR (gray) conformational changes upon EGF (orange) binding in **b** neutral and **c** partially anionic membrane. Numbers indicate electrostatic (dark red) and hydrophobic (dark green) interactions. **b** Left: the flat extracellular domain tilts the transmembrane helix, pulling the JMD (blue line) into the lipid bilayer. The negatively charged NCTT (red) interacts with the positive residues in the KD (blue) leading to the release of the CCTT. Right: with EGF bound, the upright extracellular domain allows the transmembrane helix to be vertical, causing the full JMD to protrude from the membrane. The positively-charged JMD (blue) attracts (1) the negatively-charged CTT (red). In combination with hydrophobic interactions between the CCTT and KD (2), the attraction produces an intracellular compaction. **c** Left: the transmembrane tilt and the anionic lipids embed the positively-charged JMD into the membrane, which pulls the positively-charged KD (blue) closer to (1) interact with the negatively-charged lipids (red). Right: with EGF bound, the upright transmembrane domain extends the JMD out of the membrane, further positioning the KD away from the membrane. The positive residues in the KD (blue) interact (1) with the negatively-charged CTT (red).

(Supplementary Fig. 21). The tilted TMD positions the JMD closer to the membrane surface, which increases the JMD-lipid interactions (Supplementary Figs. 22, 23). Consistently, JMD-membrane interactions have been reported in prior simulation studies[45,52]. The position and interactions of the JMD in turn influence the conformation of the entire intracellular region of the receptor.

In the neutral bilayer (Fig. 4b), EGF binding exposes the positively-charged residues of the JMD for electrostatic interactions, including with the negatively-charged residues of the CTT, leading to a more compact structure. In both JMneu and CTTneu EGFR, this interaction—and thus the EGF-dependent conformational change—are absent (Fig. 3c, Supplementary Figs. 12, 13). In the partially anionic bilayer (Fig. 4c), the positively-charged residues of the JMD tend to embed into lipid bilayer (Supplementary Fig. 22), lifting the KD and thus the intracellular domain closer to the membrane (Supplementary Figs. 18–20). In this case, the decreased tilt of the TMD with EGF bound extends the JMD, positioning the KD and subsequently the CTT further from the plasma membrane. The neutralization in JMneu EGFR likely frees the JMD from the anionic membrane, leading to an intracellular expansion similar to the case of the neutral bilayer without EGF (Fig. 3c). The neutralization in CTTneu EGFR, on the other hand, removes the repulsion between the CTT and the anionic membrane, leading to a more compact intracellular domain that still lacks the EGF-dependent effects observed in WT EGFR (Supplementary Fig. 13).

**Phosphorylation reduced by neutralization.** While ligand binding to monomeric receptors begins EGFR-mediated signaling, the subsequent changes to the conformation and organization of the receptors lead to phosphorylation of adapter proteins. To explore the correlation of the initial conformational change with downstream signaling, phosphorylation was monitored for wild-type and mutated EGFR in CHO cells in the absence and presence of EGF and epigen (Fig. 5a). JMneu EGFR showed reduced phosphorylation by 30–50% when compared to WT EGFR upon both EGF and epigen stimulation (Fig. 5b; Supplementary Tables 4, 5; Suppplementary Figs. 24, 25). Consistent with these observations, in previous studies it was found that

deletion of the charged segment of the JMD reduced phosphorylation levels by 95% and neutralization of individual residues reduced levels by up to 50%[53–55]. In addition, substitution of the native sequence with a neutral, unstructured sequence led to the disappearance of signaling, even while ligand binding and dimerization capabilities were retained[56]. The JMD has been shown to mediate autophosphorylation efficiency via conformational changes that differ with ligand identity[48,57], possibly reflecting the changes we observe here (Fig. 5c). While both reduction of phosphorylation levels in cells and a loss of extracellular/intracellular conformational coupling in vitro were seen for JMneu EGFR, interpretation of the cellular phosphorylation results is complicated by the presence of interactions with other receptors or charged proteins[44,45,55].

Phosphorylation was also monitored for the CTTneu EGFR (Supplementary Fig. 26), where a high basal level of phosphorylation was observed due to the previously established auto-inhibitory role of the mutated residues[58]. However, only a marginal increase in phosphorylation was measured upon addition of EGF, demonstrating that ligand-dependent phosphorylation also decreases relative to WT EGFR upon neutralization of the CTT. Dimerization of wild-type EGFR in the absence of ligand binding was previously shown to be insufficient for signaling, and was ascribed to an unidentified EGF-induced conformational change, such as the one identified in this study[59,60].

## Discussion

Collectively, our observations indicate that conformational changes within the intracellular region of the EGFR are dictated by the previously-reported ligand-induced conformational changes in the extracellular region. We envision that the ligand-induced and lipid-dependent conformations within the monomer precede ligand-induced oligomerization. The extracellular/intracellular conformational coupling and phosphorylation of tyrosines in the C-terminal tail bookend EGFR signaling, and as such their correlation implies that the conformational coupling may be involved in the biophysical mechanism of signaling[59]. The variable intracellular conformation may also contribute to the

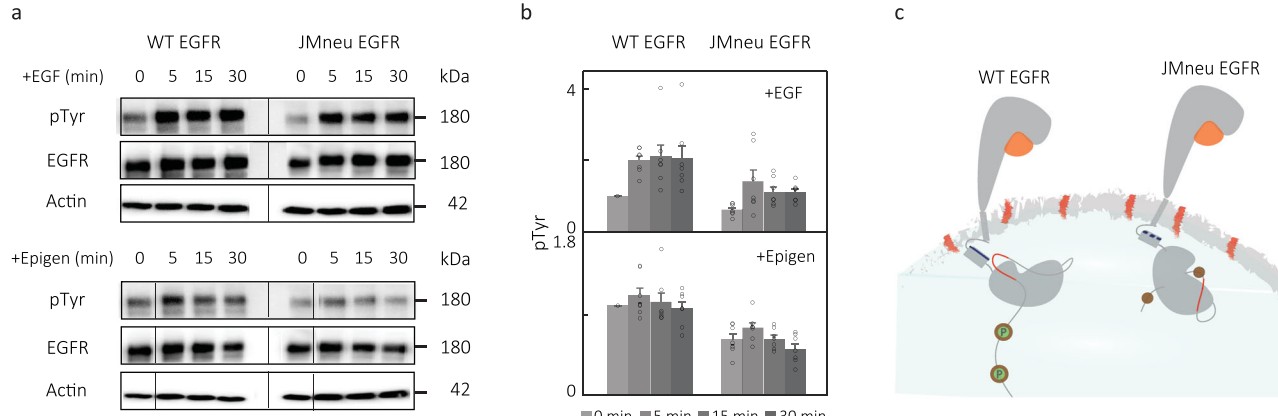

**Fig. 5 Cellular experiments with JMneu EGFR show reduced phosphorylation. a** Expression of phosphorylated EGFR and total EGFR at 0, 5, 15, and 30 min of 100 ng/mL EGF (top) and 300 ng/mL of epigen (bottom) stimulation in CHO cells transfected with WT EGFR and JMneu EGFR. Actin was used as a loading control. The blots reported are representative of seven independent biological replicates. Black line delineates boundary between gels. **b** Relative expression of phosphorylated tyrosine (pTyr) was determined by western blot quantification. The values were normalized to pTyr before EGF or epigen stimulation (time 0) of the wild-type EGFR transfected cells. Data are presented as mean values +/−SEM from seven independent biological replicates (circles); average, SEM reported in Supplementary Table 6. Two-tailed, nonparametric, paired t-test was performed to obtain the P-values (exact P-values mentioned in Supplementary Table 7). No adjustments were made for multiple comparisons. Source data are provided as a Source data file. **c** Schematic of proposed effect of JMneu EGFR on phosphorylation in CHO cells. The positive region in the JMD is shown in blue in WT EGFR and in dotted blue in JMneu EGFR. The negative lipids and region in the NCTT are shown in red. The circles in the CTT indicate tyrosine phosphorylation (P) sites.

differences in signaling efficiency and autophosphorylation site usage induced by different ligands[61,62].

Electrostatic interactions appear to define the intracellular conformation of a receptor monomer, with the positively-charged JMD as a key mediator, and so lipid charge has a profound impact on the nature of the conformations. Our results complement and expand upon previously identified electrostatic interactions between anionic lipids and the positively-charged regions of the JMD and KD, which were reported to restrict access of the substrate tyrosines to the KD and thereby regulate phosphorylation[45,48,55,63]. Similarly, ligand-induced binding of a calcium/calmodulin complex to the JMD was found to reverse the local net charge and detach the KD from the membrane, exposing the ATP binding region for substrates[44,64].

The studies presented here demonstrate that extracellular/intracellular conformational coupling is achieved through the single transmembrane helix of monomeric EGFR. In the seven transmembrane helices of rhodopsin and several other G-protein coupled receptors, highly conserved charged residues encompassing a network of electrostatic interactions are thought to lock the protein in its inactive conformation, regulating the visual transduction signaling cascade[65]. Similarly, multipass proteins have also been shown to transition into an active form through small orientational changes in the transmembrane region upon ligand binding[66,67]. Our results suggest a similar effect can be achieved within a single pass protein where one $\alpha$-helix serves as a minimal yet sufficient system for signal transduction. This system may be shared by other single pass membrane proteins with similar structures and functions[68].

## Methods

**Production of labeled full-length EGFR nanodiscs.** Fluorescently-labeled EGFR in nanodiscs was produced and characterized by adapting previous published protocols[38,39]. Plasmid encoding 6 × His-tagged version of human apolipoprotein A1 lacking the amino-terminal (ApoA1Δ49) and full-length EGFR (1210 amino acids) with a SNAP tag at the C-terminal position were codon optimized for *Escherichia coli* (*E. coli*) expression in pIVEX2.4d and SNAP-T7 vector, respectively (purchased from GenScript). Cell-free expression was carried out utilizing the Expressway Maxi Cell-Free *E. coli* Expression system (Life Technologies), which contains both ATP and the metal ion cofactors necessary for tyrosine phosphorylation, to produce full-length EGFR fused with a SNAP tag (Supplementary Fig. 2a). A final concentration of 2 mg/mL of probe sonicated 1,2-dimyristoyl-sn-glycero-3-phosphocholine (DMPC) lipid (Avanti Polar Lipids) vesicles in distilled water was added to the *E. coli* Sly D extract, in vitro protein synthesis (IVPS) *E. coli* reaction buffer, amino acids (without methoninine), methionine, T7 enzyme mix, DNA templates to make a neutral synthetic bilayer. To mimic a partially anionic symmetric bilayer, 2 mg/mL of 70% 1-palmitoyl-2-oleoyl-sn-glycero-3-phosphocholine (POPC) (Avanti Polar Lipids) probe sonicated lipid vesicles and 30% 1-palmitoyl-2-oleoyl-sn-glycero-3-phospho-L-serine (POPS) (Avanti Polar Lipids) probe sonicated lipid vesicles were used in the cell-free reaction.

A molar ratio of 500:1 lipid:Cy5 labeled 1,2-dioleoyl-sn-glycero-3-phosphoethanolamine (DOPE) lipids (Avanti Polar lipids) was added to the above solutions after bath sonication to introduce a single acceptor in the nanodisc. The lipid molar ratio was empirically optimized for one acceptor per nanodisc (Supplementary Fig. 4). 20 μg of EGFR DNA and 0.2 μg of ApoA1Δ49 DNA were added to the lysate in addition to protease inhibitor cocktail (ThermoFisher Scientific) and RNAse inhibitor (Roche). The solution was incubated at 200 rpm for 30 min at 25 °C. A feed buffer solution was made using IVPS solution, amino acids (without methionine), and methionine. A total reaction volume of 250 μL was incubated for a total time of 10 h at 200 rpm at 25 °C. Prior to the protein purification, 500 nM of snap surface 594 (New England Biolabs) was added to the reaction and was incubated at 37 °C, 150 rpm for 35 min for fluorescence labeling. Snap surface 594 is a derivative of Atto 594 with benzylguanine functionality that reacts with the genetically-encoded snap tag with near-stoichiometric efficiency[69]. For experiments measuring the distances between residues 721 and C-terminus of the protein, atto647N labeled gamma-ATP (Jena Bioscience) was added along with snap surface 594[39].

For experiments with a neutralized juxtamembrane-A (JMA) domain, JMneu EGFR was made with positively-charged residues Arg651, Lys652, Arg653, Arg656, Arg657, and Arg662 in WT EGFR mutated to neutral alanine residues. The mutations in the JMA domain are unlikely to affect helix formation or hydrophobic interactions with the membrane bilayer as the hydrophobic residues of JMA were undisturbed. Similarly, for experiments with a neutralized C-terminal tail (CTT), CTTneu EGFR was made with negatively charged residues Asp984, Asp985,

Asp988, Asp990, and Glu991 in WT EGFR mutated to neutral residues Asn984, Asn985, Asn988, Asn990, and Gln 991 (see Supplementary Fig. 10). The same protocol as above was used for cell-free expression of mutated EGFR receptors using plasmids with these modified DNA sequences.

**Affinity purification of labeled EGFR nanodiscs.** 500 μL of Ni-NTA resin slurry (Qiagen) was added to a 2 mL plastic column (Bio-Rad Laboratories). The resin was washed with double distilled water and equilibrated with 3 mL of native lysis buffer (50 mM NaH$_2$PO$_4$, 300 mM NaCl, pH 8.0). The cell-free reaction post labeling was added to 500 μL of lysis buffer on the equilibrated column and incubated at 4 °C for 2 h. The flowthrough was collected and the column was washed with lysis buffer (10 × 1 mL) followed by lysis buffer containing 10 mM imidazole (2 × 1 mL), lysis buffer containing 25 mM imidazole (2 × 1 mL), and lysis buffer containing 50 mM imidazole (2 × 1 mL) to remove all the non-specific interactions of the reaction mixture and free dye from the column. The EGFR nanodiscs were eluted with lysis buffer containing 400 mM (2 × 500 μL) imidazole. The samples were finally concentrated using 50 kDa, 500 μL spin filters (Sigma-Aldrich) by centrifugation.

**Protein content for labeled EGFR nanodiscs.** SDS-PAGE was used to confirm the production of both belt protein (at 25 kDa) and EGFR (at 160 kDa) (Supplementary Fig. 2b). Samples were mixed with 2 × Laemmli sample buffer (Bio-Rad Laboratories), 2.5% 2-mercaptoethanol (Sigma-Aldrich), and boiled for 5 min at 100 °C before running on precast stain-free gels from Bio-Rad Laboratories. Precision Plus Protein unstained standard (Bio-Rad Laboratories) marker was used for the stain-free imaging and Prestained NIR protein ladder (ThermoFisher) for fluorescence imaging. Gels were run at 170 V for 45 min. Stain-free imaging was performed on a Gel Doc imager (Bio-Rad Laboratories) and fluorescent images were acquired form Typhoon Gel FLA 9500 imager (GE Healthcare Life Sciences). The other proteins appearing in the stain-free gel are proteins not expressed completely during the cell-free reaction or the transcription and translation machinery of the cell-free reaction mixture. The specificity of snap surface 594 fluorophore binding to EGFR was confirmed through a fluorescence gel (Supplementary Fig. 2b).

**Transmission electron microscopy.** 5 μL of cell-free expressed EGFR nanodiscs in 1 × PBS buffer (137 mM NaCl, 2.7 mM KCl, 10 mM Na$_2$HPO$_4$, 1.8 mM NaH$_2$PO$_4$, pH 7.4) were added to glow-discharged carbon coated 400 mesh copper grids (Electron Microscopy Sciences) and incubated for 5 min at room temperature to allow non-specific binding of the nanodiscs to the grids. The solutions were removed by gently blotting the side of the grid with filter paper. The grids were subsequently incubated with 5 μL of 2% aqueous uranyl acetate for 30 s. Excess stain was removed similarly to the sample. The grids were air-dried, and then imaged on a FEI Tecnai transmission electron microscope (120 kV, 0.35 nm point resolution). The distribution of disc sizes was analyzed using Image J software.

**Dynamic light scattering.** The EGFR nanodiscs in 1 × PBS buffer were filtered using 0.2 μm syringe filters and their dynamic light scattering measurements were performed on a DynaPro NanoStar (Wyatt Technologies, USA). Each measurement represents an average of 50 individual runs.

**Zeta potential measurements to quantify surface charge of nanodiscs.** Titrations (0%, 30%) of negatively charged lipid (PS) in neutral lipid (PC) were performed to determine the surface charge of the nanodiscs with increasing negatively charged lipid content. Zeta potential measurements were performed on a Malvern Zetasizer Nano – ZS90 (Malvern, UK), with a backscattering detection at a constant 173° scattering angle, equipped with a 633 nm laser at 4 mW. Dip cell ZEN1002 (Malvern, UK) was used in the zeta-potential experiments. EGFR-loaded nanodiscs produced from cell-free reactions were purified as mentioned above and buffered exchanged to 0.1 × PBS. A final volume of 650 μL was prepared and transferred into the zeta dip cell. For each sample, a total of five scans, 30 runs each, with an initial equilibration time of 5 min, were recorded. All experiments were performed at 25 °C. Values of the viscosity and refractive index were set at 0.8878 cP and 1.330, respectively. Data analysis was processed using the instrumental Malvern's DTS software to obtain the mean zeta-potential value.

**Phosphorylation of EGFR in nanodiscs.** Western blot was performed using the Trans-Blot Turbo Transfer System from Bio-Rad Laboratories. The pre-loaded program for high molecular weight protein transfer was used for membrane transfer. After the transfer, the membrane was blocked in 5% non-fat dry milk (prepared in TBST buffer) for the anti-EGFR western blots, and in 5% BSA (prepared in TBST buffer) for the anti-phosphotyrosine western blots for 20 min at room temperature. The membrane was then incubated in primary antibody overnight at 4 °C. Following day, the membrane was washed and incubated with secondary antibody for 1 h at room temperature. The primary antibodies, secondary antibodies and dilutions are listed in Supplementary Table 8. The fluorescence band detection was performed using the ChemiDoc Imaging System from Bio-Rad Laboratories.

Phosphorylated tyrosines were also detected using an Y1068 antibody, which binds specifically to the phosphorylated tyrosine pY1068. 1 nM of snap surface 594 labeled EGFR nanodiscs was incubated with 0.25 nM of Y1068 antibody (alexa647 labeled or unlabeled) in the presence of 50 μM ATP, 15 mM MnCl₂, 15 mM MgCl₂, and 2 mM DTT for 30 min at room temperature. FCS experiments were performed on the mixture as described below. Ensemble FRET measurements were performed by exciting 1 nM of snap surface 594 labeled EGFR nanodiscs in the absence and increasing amounts (0.5, 2, 20 nM) of alexa 647 labeled Y1068 antibody at 550 nm. The time-resolved donor fluorescence decay was deconvolved with the instrument response function and fit to a mono-exponential.

**Preparation of ligands (EGF, neuregulin) and Cetuximab drug**. Human EGF produced in *E. coli* was purchased from Gold Biotechnology (Catalog number: 1150-04-100). Human epigen produced in *E. coli* was purchased from Peprotech. 1 μM EGF and 1 μM epigen was prepared in 1 × PBS buffer. Cetuximab produced in CHO cells was purchased from Selleckchem (Catalog number: A2000). 100 nM Cetuximab was prepared in 1 × PBS buffer. Both EGF and Cetuximab were used at concentrations higher than their dissociation constants (1 μM EGF and 100 nM Cetuximab) to ensure saturation of EGFR molecules[70,71]. Human neuregulin produced in *E. coli* was purchased from Cell Signaling Technology (Catalog number: 5218SF). 1 μM neuregulin was prepared in 1 × PBS buffer.

**Fluorescence spectroscopy**. The His-tag present on the belt protein (ApoA1Δ49) was used to immobilize the EGFR-nanodisc constructs onto the microscope coverslip *via* Ni-NTA affinity. The purified EGFR nanodiscs were diluted to ~500 pM in 1 × PBS buffer and incubated for 15 min on the Ni-NTA coated glass (from Microsurfaces, Inc.) and flushed with solution containing 2 mM 6-hydroxy-2,5,7,8-tetramethylchroman-2-carboxylic acid (Sigma-Aldrich), 25 nM protocatechuate-3,4-dioxygenase (Sigma-Aldrich), and 2.5 mM protocatechuic acid (Sigma-Aldrich). Fluorescence experiments were then carried out on a home-built confocal microscope[72]. A Ti-Sapphire laser (Vitara-S, Coherent: $\lambda_c = 800$ nm, 70 nm bandwidth, 20 fs pulse duration, 80 MHz repetition rate) was focused into a non-linear photonic crystal fiber (FemtoWhite 800, NKT Photonics) to generate a supercontinuum. Excitation light was then spectrally filtered for pulses centered at 550 or 640 nm and focused with an oil immersion objective lens (UPLSAPO100×, Olympus, NA = 1.4). Fluorescence emission was collected by the same objective and fed to the avalanche photodiodes (SPCMAQRH-15, Excelitas). A 5 μm × 5 μm area of a coverslip with immobilized receptors was scanned. Diffraction limited and spatially separated single-molecule spots were then probed individually by unblocking the laser beam to record fluorescence until photobleaching. For 550 nm excitation, fluorescence was separated using a dichroic filter SP01-561RU (Laser 2000) and passed through FF01-629/56-25 (Semrock) for donor fluorescence collection. For experiments at 640 nm, ET 645/30× (Chroma) was used as the excitation filter, FF01-629/56-25 (Semrock) as the dichroic, and FF02-685/40-25 (Semrock) for acceptor fluorescence collection. The laser power for the experiments was 2–3 μW at the sample plane.

Florescence emission was binned at 100-ms resolution to generate fluorescence intensity traces for both the donor and acceptor channels. Traces with a single photobleaching step for the donor and acceptor were considered for further analysis. Regions of constant intensity in the traces were identified by a change-point algorithm[73]. Donor traces were assigned to FRET levels until acceptor photobleaching. Consecutive bunches of 1000 photons in the donor channel were used to construct fluorescence decay curves for the FRET levels[16]. The photons were histogrammed and the distributions were fit to a mono-exponential function convolved with the instrument response function (IRF) and summed with a separately-measured background term. The fit was performed using a maximum likelihood estimator (MLE), which has been shown to be more accurate in the single-molecule regime[72,74]. The extracted lifetimes were used to construct histograms with bin sizes estimated from the square root of the total number of photon bunches.

The donor-acceptor distance (r in nm) was estimated using the following relation[40,75]:

$$r = r_o \sqrt[6]{\frac{1-E}{E}} \tag{1}$$

where $r_o$ is the calculated Förster distance (8.49 nm for snap surface 594 & cy5 dye pair)[40] and the FRET efficiency (E) being experimentally measured as:

$$E = 1 - \frac{\tau_{DA}}{\tau_D} \tag{2}$$

$\tau_{DA}$ is the fluorescence lifetime of the donor in the presence of an acceptor, and $\tau_D$ is the lifetime of donor-only construct. The distance between the donor and acceptor was quantified using a reference lifetime determined with a separately-characterized donor-only construct. For the smFRET measurements on the labeled constructs, the Cy5-lipid is confined to one side of the membrane for the duration of the measurement[76]. The 0.002% of DOPE doped with DMPC lipids in our membrane composition has a diffusion coefficient of ~$0.56 \times 10^{-8}$ cm² s⁻¹, which corresponds to a diffusion time of ~1 ms for the Cy5-lipid through the diameter of the disc[77]. Therefore, the extracted distances are a photon-weighted average over

the rapid translational diffusion of the labeled dye across the surface of the membrane nanodisc.

Previous anisotropic measurements on the intracellular domain fluorescently labeled at the C-terminus found a rotational time of 85 ns[78], with reduced dynamics of the CTT upon EGF binding[9]. The measurement time for single-molecule FRET (~100-ms), averages over this flexible motion of the CTT.

For fluorescence correlation spectroscopy (FCS) experiments, the same confocal microscope was used to probe diffusing EGFR embedded nanodisc systems at ~1 nM with the focus of the microscope objective adjusted 30 μm above the coverslip. The temporal fluctuations in the fluorescence signals were autocorrelated to generate the FCS curves, which were then analyzed using a 1-species translational diffusion model[39]:

$$G(\tau) = 1 + \frac{1}{N} \frac{1}{(1+\tau/\tau_d)\sqrt{1+s^2\,\tau/\tau_d}} \tag{3}$$

where N is the number of molecules, $\tau_d$ represents the mean residence time (set by translational diffusion), and s being the ratio of transversal to axial dimensions of the confocal volume. The translation diffusion times were then used to extract the diffusion constant D = $\omega^2/4\tau_d$, where $\omega$ represents the beam waist of the laser focus. The effective hydrodynamics radius (R) was finally derived using Stokes-Einstein relation:

$$D = \frac{k_B T}{6\pi\eta R} \tag{4}$$

where $k_B$ is the Boltzmann's constant, T being the temperature, and $\eta$ being the viscosity of the medium[79]. All experiments were performed at room temperature (21 °C) with 30% humidity. The photon arrival times were recorded by a time-correlated single-photon counting (TCSPC) module (PicoHarp 300, PicoQuant).

The change in radius (%) as a function of salt concentration in Fig. 3b was fit to the below equation following Debye-Huckel theory for electrostatic screening[50]:

$$\Delta R = ae^{-b\sqrt{c}} \tag{5}$$

where $\Delta R$ refers to the change in radius (%) and c is the salt concentration. Fit values are reported in Supplementary Table 9.

**Statistical information**. Statistical analysis was performed using MATLAB. One-way analysis of variance (ANOVA) was performed on different pairs of single-molecule FRET data and statistical significance was set at $P \leq 0.001$. The P-values, degrees of freedom, F-statistics is reported in Supplementary Table 10. The median, minimum, maximum, whiskers, and quartiles for the single-molecule FRET distributions is provided in Supplementary Tables 11, 12. The number of data points in the smFRET lifetime distributions is reported in Supplementary Table 13. Two-tailed, nonparametric, paired *t*-test was performed on different pairs of western blot phosphorylation experiments and the P-values are reported in Supplementary Tables 7, 14. The average, standard deviation, and standard error of mean for the western blot distributions is reported in Supplementary Tables 6, 15.

**Atomistic simulations with the CHARMM force field**. We utilized the ordered domain of the modeled EGFR structure (+EGF structure: residue ID: 1-995 and −EGF structure: residue ID: 3-995) to carry out microseconds long all-atom, explicit solvent simulations and characterized active and inactive EGFR embedded in the DMPC lipid bilayer. Initial configurations of these simulations were constructed using the active and inactive structures assembled in a previous work (Supplementary Fig. 27)[45]. Arkhipov et al. assembled the active dimer using the crystal structure (PDB ID: 3NJP) for the dimeric form of the extracellular domain and the crystal structure for the KD dimer (PDB ID: 2GS6). Since there are no crystal structures for the transmembrane (TM) domain, the authors modeled a TM+JM dimer based on the Her2 N-terminal dimer crystal structure (PDB ID: 2JWA) and confirmed its stability with a 100-μs long all-atom simulation. Finally, the extracellular domain, TM+JM domain, and KD were connected together. When connecting KD with the JM domain, the authors used the crystal structure 3GOP as a reference, which provides the coordinates for the dimeric KD and JM domain. We took the copy of monomer from the assembled and equilibrated active dimer structure as the initial configuration for the ligand-bound EGFR.

The unbound structure for EGFR was constructed by replacing the extracellular domain of the ligand-bound EGFR with that from an inactive configuration previously reported[45]. The inactive conformation was assembled by Shaw and coworkers using the crystal structure for the monomic extracellular domain (PDB ID: 1NQL) and the crystal structure for a single copy of inactive KD (PDB ID: 3GT8). The TM+JM domain was again predicted by molecular dynamics simulations. When connecting the extracellular domain, the TM+JM domain, and KD, the authors further rotated KD relative to the membrane to occlude the substrate-binding sites.

Both the active and chimera inactive monomers of EGFR were embedded in a neutral DMPC membrane, prepared with the CHARMM-gui toolkit[80]. Each system was simulated in a NPT ensemble for 1.5 μs with the time step of 2 fs. CHARMM36m force field[81] and TIP3P water molecules[82] were used. The systems were neutralized and solvated with 0.15 M NaCl, reaching 351,139 atoms used for the simulation of active monomer, and 294,387 atoms used for the simulation of

chimera inactive structure. The simulations were performed using GROMACS 2018[83]. The simulations reveal a closer interaction between the negatively charged N-terminal portion of the tail (NCTT) and the positively-charged juxtamembrane domain in the active EGFR, while these two domains stay far away from each other in the inactive EGFR (Supplementary Figs. 28, 29). Since the juxtamembrane domain is closer to the plasma membrane, the NCTT is closer to the membrane as well. This is consistent with the results of the coarse-grained simulations reported in the main text.

**Coarse-grained, explicit-solvent simulations with the MARTINI force field**. We carried out a set of simulations of the full-length EGFR, including the disordered CTT, using the coarse-grained MARTINI force field[51]. Though coarse-grained, these simulations provide explicit representations for the lipid bilayer, ions, water molecules, and the protein. They allow direct comparisons with the distances measured in FRET experiments. Since the Martini force field was originally calibrated for ordered proteins, when directly applied to disordered regions, the simulations produced overly collapsed configurations. Therefore, we adjusted the interaction strength between protein and water molecules to provide more reasonable conformations for the C-terminal domain. Similar modifications have been made to atomistic force fields to improve their accuracy in modeling disordered proteins[84]. Specifically, we scaled the Lennard-Jones potentials between water molecules and all protein atoms of the C-terminal domain (residues 961–1186). The scaling factor, 1.12, was chosen to best reproduce the radius of gyration of the CTT as measured by the SAXS experiment[85] (Supplementary Fig. 16). The interaction between water molecules and other parts of the EGFR was left unchanged. The time step of Martini simulation was set as 20 fs. The simulations were performed at 303 K.

With the recalibrated force field, we performed four sets of simulations of the active/inactive EGFR embedded in DMPC/POPC-POPS bilayers, respectively, for a total of 400 μs. The full-length proteins were constructed by combining the atomic models for the ordered parts (see the section "Atomistic simulations with the CHARMM force field") with an atomic model for the CTT portion of the EGFR built with Modeller[86]. We then used the CHARMM-GUI Martini Maker[87] user interface to coarse grain the full-length proteins, embed the proteins into lipid bilayers, solvate the systems with water molecules and counter ions, and generate input files for simulations with Gromacs 2019[83]. While coarse-graining the CTT, we assumed that all residues in the region adopt random coiled configurations, i.e., no specific secondary structure preference. Indeed, many experimental studies and our prior all-atom simulations support this assumption[16,85]. For random coils, the MARTINI force fields automatically assign bond length, angles, and dihedrals from a database, and no secondary structure information was extracted from the initial structure. Therefore, the initial structure does not affect equilibrium configurations explored in the coarse-grained simulations. The impact of the initial structure was further reduced with a minimization step using the coarse-grained force field before launching molecular dynamics simulations.

Umbrella sampling[88] was used to facilitate the exploration of the conformational space. The contact number between the KD and the CTT, which captures the interaction strength between these two domains, was selected as one of the collective variables. The contact number was defined as the number of atom pairs between the KD and the CTT that have a smaller than 8 Å mutual distance. The reference umbrella coordinates varied from 1000 to 4000 with a spacing of 750. To accelerate the equilibration of the system toward reference umbrella values, we used a time-dependent spring constant that increased from 0.0 to 0.0005 kcal/mol in the first 10 ns, stayed constant for the following 80 ns, and decreased linearly to 0.000005 kcal/mol in the next 10 ns. The spring constant was kept at 0.000005 kcal/mol for the remaining of the simulations. We performed an additional set of umbrella simulations that included biases over the JMA-NCTT distance, in addition to the contact number between the KD and the CTT, with the target value of 15.0 Å and the spring constant of 0.005 kcal/mol/Å$^2$.

We analyzed the simulation data using WHAM[89] to remove the effect of umbrella biases and compute the true thermodynamic averages of various metrics presented in the main text and Supplementary Figs. 17–23.

**Cell culture and reagents**. CHO cells were purchased from American Type Culture Collection (ATCC) and maintained in 10% CO$_2$ with Ham's F-12K (Kaighn's) medium (Gibco) supplemented with 10% FBS and 1% penicillin-streptomycin (both from Genessee Scientific). Antibodies recognizing the following proteins were purchased: Actin AC-15 (Sigma-Aldrich), EGFR, phospho-EGFR, phospho-EGFR Y1068, phospho-EGFR Y992, phospho-EGFR Y1045, Akt, phospho-Akt S473, p44/42 MAPK (Erk1/2), and phospho-Erk1/2 T202/Y204 (Cell Signaling Technologies). Horseradish peroxidase-conjugated goat anti-mouse and goat anti-rabbit secondary antibodies were purchased from Bio-Rad Laboratories. The primary antibodies, secondary antibodies, and dilutions are listed in Supplementary Table 16.

**Transfection**. EGFR complementary DNA (cDNA) was sub-cloned into pcDNA3.1(+). The pcDNA3.1+ JMneu EGFR and CTTneu EGFR plasmids are made by GenScript USA Inc., NJ, USA, through mutagenesis. Cells were transfected with polyethylenimine (PEI) reagent with equal amounts of each plasmid.

Transfected cells were serum-starved overnight before stimulation with 100 ng/mL epidermal growth factor (EGF) (Millipore-Sigma) and 300 ng/mL with epigen (Genscript) for 5, 15, or 30 min in serum-free medium.

**Immunoblotting**. Treated cells were collected and lysed in sample buffer (62.5 mm Tris-HCl, 2% SDS, 5% β-mercaptoethanol, 10% glycerol, 0.05% bromophenol blue). Lysates were boiled for 5 min at 95 °C, resolved by 8% SDS-PAGE, and transferred to nitrocellulose membranes, followed by immunoblotting with the indicated antibodies. Immunoblots were developed using SuperSignal West Femto Maximum Sensitivity Substrate (Thermo Scientific) on an Alpha Innotech imaging station. Band density was quantified using ImageJ (National Institutes of Health) and normalized to actin as a loading control.

**Reporting summary**. Further information on research design is available in the Nature Research Reporting Summary linked to this article.

## Data availability
The molecular dynamics simulation data generated in this study have been deposited in the zenodo database (https://doi.org/10.5281/zenodo.6564353). The following PDB files were used for the construction of the monomer for the MD simulations: 3NJP, 2GS6, 2JWA, 3GOP, 1NQL, 3GT8. Source data are provided with this paper.

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

## Acknowledgements

This work was supported by the NIH Director's New Innovator Award 1DP2GM128200-01 (to G.S.S.-C.). Research was also supported by the Laser Biomedical Research Center (NIH9P41EB015871). X.L. and B.Z. acknowledge support by startup funds from the Department of Chemistry at the Massachusetts Institute of Technology. S.D.Q. acknowledges support from the Lindemann Trust. This work was in part performed under the auspices of the U.S. Department of Energy by Lawrence Livermore National Laboratory under Contract DE-AC52-07NA27344. Research was supported in part by the National Institutes of Health under award numbers R21AI120925, R01CA155642, and R01GM117342 (to M.A.C.).

## Author contributions

S.S., R.R., K.L.C., M.A.C., and G.S.S.-C. conceived the experiments. W.H., K.L.C., and M.A.C. prepared the EGFR and ApoA1 plasmids. S.D.Q., S.S., and R.R. optimized labeled EGFR production. S.S. and R.R. performed and analyzed the fluorescence experiments. X.L. and B.Z. designed and performed the simulations. C.A.D. performed the cell culture experiments. X.C. performed the EGFR function-related and characterization experiments. S.S., R.R., and G.S.S.-C. co-wrote the manuscript. All authors discussed the results and commented on the manuscript.

## Competing interests

The authors declare no competing interests.
