## [Peer Review File · Nature Communications]

REVIEWER COMMENTS

Reviewer #1 (Remarks to the Author):

In this paper, the authors show that ligand binding to the extracellular domain of monomeric EGFR leads to conformational changes in EGFR intracellular domain. They use single molecule FRET to study monomeric EGFR in nanodisks, where one of the FRET dyes is at the C-terminus of the protein, and the other is on a labeled lipid in the nanodisk. They measure the fluorescence lifetime of the donor at the C-terminus of the receptor, with and without ligand and build lifetime histograms. They see differences in the lifetimes. This suggests that there are differences in the average distance between the EGFR C-terminus and the lipids in the disk, in the presence and absence of ligand.

While the FRET measurements appear carefully done, I do not understand what the premise behind this work is, and what the significance of the findings are. EGFR monomers are inactive. Only EGFR dimers are active, and the EGFR dimer structure is not determined exclusively by the monomer structure, as contacts between the two chains in the dimer will be contributing significantly to the dimer structure. As such, the behavior of the EGFR monomer that the authors study is not directly related to EGFR function.

The mutagenesis in Figure 4 can only show that the sequence alterations are important for the function of EGFR dimers, not monomers. In no way can the findings in Figure 4 be directly correlated with the conformation of the EGFR monomer, i.e with the data in Figures 1, 2, 3.

Reviewer #2 (Remarks to the Author):

The paper titled "Ligand-induced transmembrane conformational coupling in monomeric EGFR": In this work the authors have investigated the possible role of monomer EGFR in the presence and absence of ligand, especially conformational changes occurring over the intracellular regions. A lot of work has been done on specific domains of receptor tyrosine kinases, and very few studies have highlighted the role of ligand in mediating the changes towards the intracellular disordered regions. Also interesting to see the complementarity between the experimental and simulations results.

Nevertheless, I believe the study performed is a good indicator for studying the role of full-length EGFR receptor conformational dynamics and deserves publication with some minor changes/addressing the comments:

1. In the section "Simulations capture measured distance" the author mention simulations were carried out on active(+EGF) or inactive (-EGF) conformations. Usually, from studies, it has been dictated the EGFR protein gets activated by ligand binding and dimerization process. It would be nice to mention a line on what basis the author refers to the protein as an active and inactive state? Any particular arrangement of the extracellular/intracellular region when the EGF is bound? Maybe even have a figure in supplementary showing the different conformation of active and inactive. Supp Fig 24 doesn't give more understanding on the same.

2. The 1.5 microsecond is a good amount of simulation time in all-atom to understand the conformational dynamics. But authors could have replicas of the simulations instead of running one long simulation for better statistical quantitative estimates.

3. Another receptor belonging to the same protein family such as ErbB2 has no ligand known to be bound for the activation process. But ErbB2 receptors are highly seen in breast cancers, can the author comment on this more?

4. The choice of the lipid membrane composition needs to be made clear, together with a brief discussion of how it could influence results.

5. The abstract contains a typo: "Wtih"

6. On binding of EGF the JM and CTT regions are exposed towards lipids for interaction in partially anionic lipids. Authors could highlight the residues involved and these interactions for better clarity. There are already some literature based on the same. Author could highlight and compare and give a sentence or two mentioning how similar the data is or if different from literature why do they see the difference?

7. In methods, Coarse grain model: the active conformer was modeled from the crystal structure. Author hasn't mentioned about the which X-ray crystal structure has been taken. Please provide the PDB ID.

8. In line with the above comment : author hasn't provided clear details on which active or inactive structures have been taken from PDB as their starting structure? From the information provided it seems the extracellular domain has been swapped for inactive, please make it more clear for better clarity.

9. Author has performed substantial amount of coarse grain simulations, they could have taken the conformers from the same and then backmapped to atomistic structure and simulated them? This would have become more meaningful other than comparing the same procedure in different ways. Author can provide a statement for the same.

10. A suggestion and a comment: Author have modeled the CTT region using Modeller, since already its a disordered region, authors could have minimized this structure before moving on to Coarse grain them. So as to have a better stable conformer as a starting structure.

Reviewer #3 (Remarks to the Author):

The question of how cells interpret the extracellular environment (i.e. messages from other cells) is a central question in biology. Cell membrane receptors provide the first line of communication in this signalling cascade but exactly how ligand binding to the extracellular portion of a receptor transmits information to the other side of the membrane is still an open, unresolved question.

The EGFR is arguably one of the most studied single pass membrane receptors because of its central role in normal physiology but also because of its links to disease states such as cancer. The continued development of therapeutics against this important drug target attest to its significance in our fight against cancer.

At an academic level the last couple of decades has seen a plethora of atomic level models of fragments of the EGFR. While these fragment studies have given much detail on how ligand binding transmits conformational changes in the extracellular domain which then lead to extracellular domain dimerization, the issue of how these processes couple in a full length receptor in a cell membrane environment remain to be fully delineated. On the other hand, studies in living cells, have revealed the existence of monomers, dimers higher-order oligomeric activated states of the receptor molecule. The superposition of a multitude of receptor conformational, oligomeric, and

activation states makes identification of key processes difficult. We need to go back to the drawing board, as it were.

The paper presented here addresses an important gap in our understanding of how the EGFR works. By eliminating the receptor-receptor part of the process, the authors can study how ligand binding to the extracellular domain alters the conformation of the intracellular domains. The authors specifically use lipid nanodisks and single molecule spectroscopy to optically identify single monomeric receptors. Time-resolved spectroscopy-based FRET between a c-terminal donor probe and a membrane-resident acceptor probe is employed to measure donor-acceptor distance distributions, which reflect the conformation of the intracellular domain. Lifetime-based FRET determinations are robust since they are based on a kinetic measurement not an intensity. Significantly, the authors reveal changes in the donor-acceptor distance distributions consistent with expansion or compaction of the intracellular domains, dependent on absence of ligand, presence of ligand or ligand plus a ligand-blocking antibody. The authors also reveal the significance of electrostatic interactions by comparing neutral with charged lipid surfaces, and neutralizing the charges in the juxta-membrane domain or the cytoplasmic domain. The elegant experimental studies are complemented with molecular dynamics simulations and studies with mutant receptors in living cells. Overall, the picture of a mechano-electrostatic coupling mechanism involving changes in the extracellular domain after ligand binding, transmitted through the transmembrane helix, JMD and cytoplasmic domain appears to be consistent with experiments and related literature.

I have some questions which mainly arise from my own curiosity and do not necessarily imply that the work has any major problems.

- (1) The acceptor probe is imbedded in the membrane and is free to move laterally. What effect, if any, does the movement of the acceptor probe have on the distance measured ?
- (2) What is the effect of any flexibility of the C-terminal tail on the observed FRET measurements. On what time scale does the c-terminal tail move ? Does the width of the lifetime histograms reflect any flexibility ?
- (3) Can you clarify whether the studies were conducted in the presence of ATP ? Is the C-terminal tail phosphorylated in these experiments ? And if it is, to what extent ?
- (4) Related to (3), have you considered making mutations in the C-terminal tail (replace the tyrosine(s) with appropriately changes amino acids to mimic phosphorylation)
- (5) Some of the earlier biophysical studies on the isolated intracellular domains seem to be missing from the reference list (the work by John Koland, for example)
- (6) From an allosteric view point do the data indicate an induced conformation produced by ligand or a selected conformation from a pre-existing ensemble of conformations ?

Response to Reviewers

We thank and appreciate the time and consideration of the reviewers. Our comments are in blue, and all changes to the text are marked in orange in the responses and the manuscript. We have added one additional figure in the main text and five additional figures to the supplementary information of the manuscript. The page/line numbers, figure numbers, and reference numbers provided in our response follow the numbering scheme in the revised manuscript.

Reviewer # 1:

In this paper, the authors show that ligand binding to the extracellular domain of monomeric EGFR leads to conformational changes in EGFR intracellular domain. They use single molecule FRET to study monomeric EGFR in nanodisks, where one of the FRET dyes is at the C-terminus of the protein, and the other is on a labeled lipid in the nanodisk. They measure the fluorescence lifetime of the donor at the C-terminus of the receptor, with and without ligand and build lifetime histograms. They see differences in the lifetimes. This suggests that there are differences in the average distance between the EGFR C-terminus and the lipids in the disk, in the presence and absence of ligand.

Comment 1: While the FRET measurements appear carefully done, I do not understand what the premise behind this work is, and what the significance of the findings are. EGFR monomers are inactive. Only EGFR dimers are active, and the EGFR dimer structure is not determined exclusively by the monomer structure, as contacts between the two chains in the dimer will be contributing significantly to the dimer structure. As such, the behavior of the EGFR monomer that the authors study is not directly related to EGFR function.

Response 1: We thank the reviewer for their acknowledgement of the high level of our experiments and apologize for our lack of clarity as to the significance. We address this lack through the following major revisions:

(1) Within EGFR signaling, ligands bind to EGFR monomers, as the receptors are predominantly found in their monomeric form. Thus, our studies characterize the first step of EGFR signaling, regardless of downstream oligomerization state [Li, *Cancer Cell* (2005)]. To clarify this point, we have made the following additions:

Abstract, page 2, line 6: Single pass cell surface receptors regulate cellular processes by transmitting ligand-encoded signals across the plasma membrane via changes to their extracellular and intracellular conformations. This transmembrane signaling is generally initiated by ligand binding to the receptors in their monomeric form. While subsequent receptor-receptor interactions are key aspects of transmembrane signaling, the contribution of the single transmembrane helix of monomeric receptors has been challenging to isolate due to the complexity and ligand-dependence of these interactions.

Introduction, page 2, line 25: Their signaling pathways are primarily initiated by extracellular ligand binding to monomeric receptors, which causes intracellular autophosphorylation and subsequent recruitment of adaptor proteins to the phosphorylated residues¹.

Introduction, page 3, line 43: Prior to ligand binding, 95% of EGFR is found in its monomeric form in cells²². While EGF-induced dimers have been long established as an active form of the receptors, emerging evidence suggests that other oligomerization states of EGFR also play a role in phosphorylation and signaling.

Introduction, page 3, line 52: Consistently, no homodimerization was observed for the ligand epigen, yet it still induces signaling³⁵⁻³⁷. Despite these multiple lines of evidence, the behavior of monomeric EGFR prior to oligomerization and its contribution, if any, to the signaling pathway have not yet been determined.

(2) We appreciate that no dimers are present in our work, and that the interactions within the dimer structure will lead to additional conformational changes. Having said this, emerging evidence indicates that EGFR signaling involves many oligomerization states. These results include: (i) participation of monomers in EGF-induced signaling [Koland, *et al.*, *JBC* (1988)]; (ii) an expanded dynamic range via multimer formation [Needham, *et al.*, *Nat. Commun.* (2016)]; (iii) discovery of epigen, a natural ligand that induces signaling yet not homodimerization [Freed, *et al.*, *Cell* (2017)]; and (iv) dimer formation is insufficient for signaling, leading to a proposal that conformational changes, such as those reported here, are required [Liang, *et al.*, *Cell Rep.* (2018)].

To clarify these points and establish the generality of our observations, we expanded our study to include epigen, which belongs to the EGF family yet does not induce homodimerization. While the structure of epigen-induced oligomers, if any, remains an open question, the lack of homodimerization establishes that the nature of any oligomers is distinct from EGF-induced dimers. As described below, similar results were observed with epigen as with EGF. The reviewer's comments paved the way for this addition in the revised manuscript, and are very much appreciated.

Results, page 7, line 131: To establish the generality of the extracellular/intracellular conformational coupling of EGFR, we investigated the effect of epigen, a ligand that belongs to the EGF family yet does not induce homodimerization. In the presence of 1 μ M epigen, the lifetime distribution peaked at \sim 1.5 ns (8.5 nm), indicating an intracellular compaction was observed for both bilayers (figure 2a and d, bottom). While the specific conformational response is ligand dependent^{8,48}, these distributions establish that the extracellular/intracellular conformational coupling observed through smFRET is not uniquely induced by EGF.

Figure 2:...(a) The donor fluorescence lifetime distribution in DMPC bilayer...with 1 μ M epigen (bottom). (d) The donor fluorescence lifetime distribution in POPC-POPS bilayer...with 1 μ M epigen (bottom).

Results, page 9, line 149: We performed smFRET measurements on both mutants in neutral and partially anionic lipid bilayers and in the presence and absence of EGF and of epigen. We built donor lifetime histograms for the ten samples (Fig. 3c and Supplementary Fig. 12, 13). In contrast to the distributions for WT EGFR (Fig. 2 a, d), the distributions for the mutant samples were statistically similar in the absence and presence of EGF and of epigen in both bilayers (Supplementary Fig. 14, 15), indicating that the ligand-induced intracellular conformational changes are suppressed upon mutation.

Figure 3:...(c) smFRET donor fluorescence lifetime histograms for JMneu EGFR in POPC-POPS bilayer without ligand (top).....with 1 μ M epigen (bottom).

The additional statistical analysis of these distributions is described in Supplementary Figures 7, 9, and 14 and Table 3, 7 and 8.

Consistently, the level of phosphorylation upon epigen stimulation decreased for JMneu EGFR as compared to WT EGFR in CHO cells. We describe the phosphorylation results with the following additions to the manuscript:

Results, page 13, line 219: To explore the correlation of the initial conformational change with downstream signaling, phosphorylation was monitored for wild-type and mutated EGFR in CHO

cells in the absence and presence of EGF and epigen (Fig. 5a). JMneu EGFR showed reduced phosphorylation by 30%-50% when compared to WT EGFR upon both EGF and epigen stimulation (Fig. 5b; Supplementary Table 4, 5; Supplementary Fig. 24, 25).

Figure 5: (a) Expression of phosphorylated EGFR and total EGFR at 0, 5, 15, and 30 minutes of 100~ng/mL EGF (top) and 300~ng/mL of epigen (bottom) stimulation in CHO cells transfected with WT EGFR and JMneu EGFR. (b) Relative expression of phosphorylated tyrosine (pTyr) was determined by western blot quantification. The values were normalized to pTyr before EGF or epigen stimulation.

The additional data from these measurements is in Supplementary Figure 24 and Supplementary Table 4.

Comment 2: The mutagenesis in Figure 4 can only show that the sequence alterations are important for the function of EGFR dimers, not monomers. In no way can the findings in Figure 4 be directly correlated with the conformation of the EGFR monomer, i.e with the data in Figures 1, 2, 3.

Response 2: We thank the reviewer for raising this concern and again apologize for our lack of clarity. As mentioned above, even within the dimer model, our study bookends EGFR signaling. That is, here we probe the first step, ligand binding (smFRET experiments, Figs. 1-4), and the final step, phosphorylation (cell experiments, Fig. 5), of the overall signaling pathway. While many in vivo interactions and components are missing, we investigate these two steps to identify effects that may span EGFR-mediated signaling. We have revised the text to better describe this relationship as follows:

Results, page 13, line 218: While ligand binding to monomeric receptors begins EGFR-mediated signaling, the subsequent changes to the conformation and organization of the receptors lead to phosphorylation of adaptor proteins. To explore the correlation of the initial conformational change with downstream signaling, phosphorylation was monitored for wild-type and mutated EGFR in CHO cells in the absence and presence of EGF and epigen (Fig. 5a).

Discussion, page 14, line 246: We envision that the ligand-induced and lipid-dependent conformations within the monomer precede ligand-induced oligomerization. The extracellular/intracellular conformational coupling and phosphorylation of tyrosines in the C-terminal tail bookend EGFR signaling, and as such their correlation implies that the conformational coupling may be involved the biophysical mechanism of signaling⁵⁹.

Reviewer # 2:

The paper titled “Ligand-induced transmembrane conformational coupling in monomeric EGFR”: In this work the authors have investigated the possible role of monomer EGFR in the presence and absence of ligand, especially conformational changes occurring over the intracellular regions. A lot of work has been done on specific domains of receptor tyrosine kinases, and very few studies have highlighted the role of ligand in mediating the changes towards the intracellular disordered regions. Also interesting to see the complementarity between the experimental and simulations results.

Nevertheless, I believe the study performed is a good indicator for studying the role of full-length EGFR receptor conformational dynamics and deserves publication with some minor changes/addressing the comments:

Response: We appreciate the reviewer's strong assessment of the significance, novelty, and broad interest of the paper. We also thank them for the helpful suggestions and comments that have improved the clarity and quality of our manuscript.

Comment 1: In the section “Simulations capture measured distance” the author mention simulations were carried out on active (+EGF) or inactive (-EGF) conformations. Usually, from studies, it has been dictated the EGFR protein gets activated by ligand binding and dimerization process. It would be nice to mention a line on what basis the author refers to the protein as an active and inactive state? Any particular arrangement of the extracellular/intracellular region when the EGF is bound? Maybe even have a figure in supplementary showing the different conformation of active and inactive. Supp Fig 24 doesn't give more understanding on the same.

Response 1: We apologize for our lack of clarity. We changed the nomenclature to the more specific ligand-bound (+EGF) and ligand-unbound (-EGF) conformations. Following the reviewer's suggestion, we also added a figure (shown below) to highlight the differences in the conformations. As is now evident from the figure, the two initial structures differ significantly in the extracellular domain, with the ligand-bound structure featuring a more upright configuration compared to the one in the unbound form. The conformations for the transmembrane and intracellular domains are identical by construction.

Supplementary Fig. 27: Structural comparison between the ligand bound (+EGF, red) and unbound (-EGF, blue) EGFR. The two structures were aligned based on the intracellular kinase domain. The disordered CTT is not resolved in the crystal structure and is not shown in this figure. The lipid bilayer is shown in gray. These modeled structures were used to initialize our simulations with the CHARMM36m force field (CTT excluded) and the MARTINI force field (CTT included). The extracellular domain in the +EGF structure adopts a more upright configuration than the -EGF structure. The transmembrane and intracellular domains are the same between these two structures by construction.

Comment 2: The 1.5 microsecond is a good amount of simulation time in all-atom to understand the conformational dynamics. But authors could have replicas of the simulations instead of running one long simulation for better statistical quantitative estimates.

Response 2: We thank the reviewer for this helpful suggestion to improve the rigor of our conclusions. To establish the statistical significance of our results, we performed two additional one-microsecond-long all-atom simulations for the EGF unbound and bound EGFR. These simulations started from the same initial configurations used to initiate the two simulations presented in Supplementary Fig. 28 but with different random seeds and atomic velocities. All trends were qualitatively reproduced. As illustrated in the new Supplementary Fig. 29 (shown below), the distance between the N-terminal portion of the tail (NCTT) and the juxtamembrane domain in all three simulations is shorter in the ligand-bound (+EGF) EGFR than the unbound one (-EGF).

Supplementary Fig. 29: Distributions of JMA-NCTT distance from three-independent all-atom simulations. Details of these simulations are provided in the Section: *Atomistic Simulations with the CHARMM Force Field*. All simulations support a closer distance between the NCTT and JMA in the +EGF EGFR.

Comment 3: Another receptor belonging to the same protein family such as ErbB2 has no ligand known to be bound for the activation process. But ErbB2 receptors are highly seen in breast cancers, can the author comment on this more?

Response 3: We thank the reviewer for highlighting this interesting aspect of EGFR signaling. As the reviewer mentions, the different members of the EGFR family play distinct roles in cellular signaling and disease. The unique open conformation of the ErbB2 extracellular domain does not bind ligands but makes ErbB2 the preferred binding partner of all ErbB receptors [Cho, *Nature* (2003)]. Dimerization with ErbB2 results in increased ligand affinity of the other protomer, increased coupling efficiency to signaling molecules, and decreased rate of receptor internalization [Karunakaran, *EMBO J.* (1996)]. All of these effects are enhanced in breast cancers [Tan, *Breast Cancer Sensitivity* (2007)]. Therefore, ErbB2 receptors employ distinct mechanisms from EGFR, and we hope to perform comparative studies of these conformations in the future. To clarify this interesting point, we have added the following:

Introduction, page 3, line 46: Both homo- and heterodimerization between members of the EGFR family have been observed^{25,26}. The nature of the dimer can enhance ligand affinity or protein binding, providing an alternative mechanism to control signaling levels²⁷⁻³⁰.

Comment 4: The choice of the lipid membrane composition needs to be made clear, together with a brief discussion of how it could influence results.

Response 4: We apologize for our lack of clarity. We have improved our explanation of the membrane composition and its potential effects with the following additions to the manuscript:

Results, page 4, line 76: To first benchmark the behavior, monomeric wild type (WT) EGFR was embedded in a nanodisc with a 1,2-Dimyristoyl-sn-glycero-3-phosphocholine (DMPC) bilayer, which is neutrally charged and thus lacks the complex electrostatic interactions of the in vivo plasma membrane.

Results, page 6, line 110: While the neutral bilayer provided the simplest environment for initial experiments, we then ascertained the effect of a near-native lipid composition on the intracellular region by incorporating a partially anionic bilayer into the nanodiscs (70% 1-palmitoyl-2-oleoyl-sn-glycero-3-phosphocholine, POPC; 30% 1-palmitoyl-2-oleoyl-sn-glycero-3-phospho-L-serine, POPS; Fig. 2c, Supplementary Fig. 3), which replicates the plasma membrane anionic lipid content of mammalian cells⁴³. The electrostatic interactions introduced by these anionic lipids replicates this important aspect of the cellular environment, which have been previously implicated in the regulation of EGFR signaling^{44,45}.

Comment 5: The abstract contains a typo: “Wtih”

Response 5: We have corrected the typo in the revised manuscript, and thank the reviewer for bringing it to our attention.

Comment 6: On binding of EGF the JM and CTT regions are exposed towards lipids for interaction in partially anionic lipids. Authors could highlight the residues involved and these interactions for better

clarity. There are already some literature based on the same. Author could highlight and compare and give a sentence or two mentioning how similar the data is or if different from literature why do they see the difference?

Response 6: Following the reviewers' suggestion, we examined the interactions between the JM and the lipid bilayer and found that our simulations produced results consistent with previous computational studies [Arkhipov, *Cell* (2013); Halim, *Biochim. Biophys. Acta* (2015)]. We thank the reviewer for highlighting this additional parameter as a tool to improve the comparison of our results with the literature.

Specifically, Arkhipov, *et al.* reported that part of the monomeric JMA (residues 656-662) can form close contacts with an anionic membrane. Halim, *et al.* further examined JMA-membrane interactions by separately analyzing the JMA-membrane distance for residues 645-647 and for residues 655-659 of the JMA. Their simulations found that JMA-membrane interactions are primarily due to residues 645-647. Replicating this analysis on our simulations reproduced this result, as shown through the additions below:

Results, page 11, line 199: The tilted TMD positions the JMD closer to the membrane surface, which increases the JMD-lipid interactions (Supplementary Fig. 22, 23). Consistently, JMD-membrane interactions have been reported in prior simulation studies^{45, 52}.

Supplementary Fig. 23: Close contacts between JMA and the membrane revealed by coarse-grained simulations. (a) Probability distributions of the minimum distance between two regions of the JMA domain (R645-647, R655-659) and the phosphate groups of the inner leaflet of the bilayer. The median values of the distributions are indicated by vertical lines. The shaded regions represent the standard deviation estimated with block averaging. (b) A representative structure of the ligand-bound EGFR from simulations with the POPC-POPS bilayer highlighting the interactions between the JMA domain (R645-647 (blue), R655-659 (cyan)) and the phosphate groups. The inset provides a closer look of the contacts.

Comment 7: In methods, Coarse grain model: the active conformer was modeled from the crystal structure. Author hasn't mentioned about the which X-ray crystal structure has been taken. Please provide the PDB ID.

Response 7: We apologize for this missing information. Both +EGF and -EGF conformers were taken from the modeled structure of a previous study in which the complete EGFR structure was modeled based on a series of solved crystal structures of EGFR components [Arkhipov, *Cell* (2013)]. We have revised the manuscript to clarify these details by including the following text in the *Section: Atomistic Simulations with the CHARMM Force Field* in the supplementary information.

Initial configurations of these simulations were constructed using the active and inactive structures assembled in a previous work (Supplementary Fig. 27)¹⁸. Arkhipov *et al.* assembled the active dimer using the crystal structure (PDB ID: 3NJP) for the dimeric form of the extracellular domain and the crystal structure for the KD dimer (PDB ID: 2GS6). Since there are no crystal structures for the transmembrane (TM) domain, the authors modeled a TM+JM dimer based on the Her2 N-terminal dimer crystal structure (PDB ID: 2JWA) and confirmed its stability with a 100- μ s-long all-atom simulation. Finally, the extracellular domain, TM+JM domain, and KD were connected together. When connecting KD with the JM domain, the authors used the crystal structure 3GOP as a reference, which provides the coordinates for the dimeric KD and JM domain. We took the copy of monomer from the assembled and equilibrated active dimer structure as the initial configuration for the ligand-bound EGFR.

The unbound structure for EGFR was constructed by replacing the extracellular domain of the ligand-bound EGFR with that from an inactive configuration previously reported¹⁸. The inactive conformation was assembled by Shaw and coworkers using the crystal structure for the monomeric extracellular domain (PDB ID: 1NQL) and the crystal structure for a single copy of inactive KD (PDB ID: 3GT8). The TM+JM domain was again predicted by molecular dynamics simulations. When connecting the extracellular domain, the TM+JM domain, and KD, the authors further rotated KD relative to the membrane to occlude the substrate-binding sites.

We also provided more information in the *Section: Coarse-grained, Explicit-solvent Simulations with the MARTINI Force Field* in the supplementary information to better explain the construction of initial configurations used in the coarse-grained simulations. The newly added text is:

The full-length proteins were constructed by combining the atomic models for the ordered parts (see the *Section: Atomistic Simulations with the CHARMM Force Field*) with an atomic model for the CTT portion of the EGFR built with Modeller²⁶. We then used the CHARMM-GUI Martini Maker²⁷ user interface to coarse grain the full-length proteins, embed the proteins into lipid bilayers, solvate the systems with water molecules and counter ions, and generate input files for simulations with Gromacs 2019²².

Comment 8: In line with the above comment : author hasn't provided clear details on which active or inactive structures have been taken from PDB as their starting structure? From the information provided it seems the extracellular domain has been swapped for inactive, please make it more clear for better clarity.

Response 8: We apologize for this missing information again. As detailed in the *Response to Comment 7*, we have revised the text in the supporting information to include detailed procedures used to construct the

two initial monomeric structures of ligand-bound and unbound EGFR. We have also added a Supplementary Figure 27, as shown in the *Response to Comment 1*, to illustrate these two structures.

Comment 9: Author has performed substantial amount of coarse grain simulations, they could have taken the conformers from the same and then backmapped to atomistic structure and simulated them? This would have become more meaningful other than comparing the same procedure in different ways. Author can provide a statement for the same.

Response 9: We thank the reviewer for this suggestion. We carried out the back mapping procedure suggested by the reviewer to evaluate the stability of key configurations from coarse-grained simulations with the all-atom CHARMM36m force field. From this procedure, we found good agreement for several parameters implicated in the transmembrane conformational coupling and in determining the intracellular conformation.

Specifically, we selected four representative structures from the MARTINI simulations of the four systems (+EGF or -EGF EGFR embedded in DMPC or POPC-POPS lipids). These structures exhibit order parameters comparable to the values presented in Figure 4a. They were identified from a single-linkage clustering algorithm over the simulated EGFR configurations using the RMSD of Ca atoms. These protein structures and their associated MARTINI lipid bilayer were back mapped into an atomistic structure before resolvating the system with water molecules and counter ions for all-atom simulations. Starting from the back mapped configurations, we performed energy minimization of the whole system, followed by 1.875-ns-long simulations during which we gradually reduced the harmonic positional restraints applied on protein and lipid atoms. We performed additional 20-ns-long equilibration simulations with protein atoms being restricted to starting positions while the rest of the system was allowed to relax freely. After equilibration, we carried out production runs lasting over 125 ns for each system.

The atomistic simulations reproduced the trends observed for TMD tilting angles between ligand-bound (+EGF) and unbound (-EGF) systems (Supplementary Fig. 21), KD-CCTT contacts (Supplementary Fig. 20), and JMA-membrane contacts (Supplementary Fig. 22, 23). In the simulations, these parameters were identified as responsible for key aspects of the observed behaviors, including: the TMD tilting angle transduced the ligand-binding event across the bilayer; the KD-CCTT contacts maintained the distance between the C-terminus of the CTT and the lipid bilayer, which is the measured distance; and the JMA is more embedded in the POPC-POPS bilayer than in the DMPC bilayer, which is the origin of the measured lipid dependence.

While the majority of the trends were produced (Figure R1a-c), the trend for the NCTT-bilayer distance was not consistent between the all-atom (Figure R2d) and coarse-grained simulations (Supplementary Fig. 19). This inconsistency is likely due to one or both of the following factors: (1) the short simulation time of 125 ns for all-atom simulations as compared with MARTINI simulations (100 μ s for each system). The statistical sampling of different protein configurations is presumably less converged, especially considering the large fluctuation for this distance and the small differences for the four samples; and (2) we observed a collapse of the CTT during all-atom simulations, and the simulated radius of gyration for CTT is smaller than the experimental value (Figure R2). Such effects are well-known for

simulations of disordered regions [Best, *J. Chem. Theory Comput.* (2015)], and may have hampered our ability to reproduce this result from simulations with the MARTINI force fields, as we adjusted the force field for the original coarse-grained simulations to ensure accurate modeling of the CTT.

Figure R1: Probability distributions of different variables from the backmapped atomistic simulations for ligand-bound (orange) and unbound (purple) EGFR embedded in the DMPC (top) and POPC-POPS (bottom) bilayer. The vertical lines correspond to ensemble averages. The results for KD-CCTT contacts (a), the TMD tilting angle (b), and JMA-membrane contacts (c) are consistent with those obtained from the MARTINI force fields. However, the simulations fail to resolve the subtle differences among the NCTT-membrane distance (d) across the four systems, presumably due to their short length.

Figure R2. Probability distribution of the radius of gyration (R_g) of the CTT (residues 961-1186) calculated from the four backmapped atomistic simulations. The vertical line corresponds to the ensemble average. The average R_g is much smaller than 4.66 nm, the value reported by the SAXS experiment [Keppel, *J. Biol. Chem.* (2017)] and that calculated from our coarse-grained simulations (Supplementary Fig. 16). Therefore, the state-of-the-art atomistic force fields (CHARMM36m) still overly compact the disordered CTT region.

Comment 10: A suggestion and a comment: Author have modeled the CTT region using Modeller, since already its a disordered region, authors could have minimized this structure before moving on to Coarse grain them. So as to have a better stable conformer as a starting structure.

Response 10: We thank the reviewer for this insightful comment. We have added the text below to explain the procedure by which a stable initial structure was constructed. We also agree that it would be interesting to explore the role of residual secondary structures in the future with a better prepared initial structure, perhaps using predictions from bioinformatics tools such as RaptorX [Wang, *Nucleic Acids Res.* (2016)]. The newly added text is:

Supporting Information, Section *Coarse-grained, Explicit-solvent Simulations with the MARTINI Force Field*: While coarse-graining the CTT, we assumed that all residues in the region adopt random coiled configurations, i.e., no specific secondary structure preference. Indeed, many experimental studies and our prior all-atom simulations support this assumption^{8,25}. For random coils, the MARTINI force fields automatically assign bond length, angles, and dihedrals from a database, and no secondary structure information was extracted from the initial structure. Therefore, the initial structure does not affect equilibrium configurations explored in the coarse-grained simulations. The impact of the initial structure was further reduced with a minimization step using the coarse-grained force field before launching molecular dynamics simulations.

Reviewer # 3:

The question of how cells interpret the extracellular environment (i.e. messages from other cells) is a central question in biology. Cell membrane receptors provide the first line of communication in this signalling cascade but exactly how ligand binding to the extracellular portion of a receptor transmits information to the other side of the membrane is still an open, unresolved question.

The EGFR is arguably one of the most studied single pass membrane receptors because of its central role in normal physiology but also because of its links to disease states such as cancer. The continued development of therapeutics against this important drug target attest to its significance in our fight against cancer.

At an academic level the last couple of decades has seen a plethora of atomic level models of fragments of the EGFR. While these fragment studies have given much detail on how ligand binding transmits conformational changes in the extracellular domain which then lead to extracellular domain dimerization, the issue of how these processes couple in a full length receptor in a cell membrane environment remain to be fully delineated. On the other hand, studies in living cells, have revealed the existence of monomers, dimers higher-order oligomeric activated states of the receptor molecule. The superposition of a multitude of receptor conformational, oligomeric, and activation states makes identification of key processes difficult. We need to go back to the drawing board, as it were.

The paper presented here addresses an important gap in our understanding of how the EGFR works. By eliminating the receptor-receptor part of the process, the authors can study how ligand binding to the extracellular domain alters the conformation of the intracellular domains. The authors specifically use

lipid nanodisks and single molecule spectroscopy to optically identify single monomeric receptors. Time-resolved spectroscopy-based FRET between a c-terminal donor probe and a membrane-resident acceptor probe is employed to measure donor-acceptor distance distributions, which reflect the conformation of the intracellular domain. Lifetime-based FRET determinations are robust since they are based on a kinetic measurement not an intensity. Significantly, the authors reveal changes in the donor-acceptor distance distributions consistent with expansion or compaction of the intracellular domains, dependent on absence of ligand, presence of ligand or ligand plus a ligand-blocking antibody. The authors also reveal the significance of electrostatic interactions by comparing neutral with charged lipid surfaces, and neutralizing the charges in the juxta-membrane domain or the cytoplasmic domain. The elegant experimental studies are complemented with molecular dynamics simulations and studies with mutant receptors in living cells. Overall, the picture of a mechano-electrostatic coupling mechanism involving changes in the extracellular domain after ligand binding, transmitted through the transmembrane helix, JMD and cytoplasmic domain appears to be consistent with experiments and related literature.

We thank the reviewer for their positive comments regarding the impact and insights of our study.

I have some questions which mainly arise from my own curiosity and do not necessarily imply that the work has any major problems.

Comment 1: The acceptor probe is imbedded in the membrane and is free to move laterally. What effect, if any, does the movement of the acceptor probe have on the distance measured?

Response 1: We thank the reviewer for this question. The labeled dye diffuses across the surface of the membrane nanodisc on a microsecond timescale [Hsieh, *J. Phys. Chem. B* (2014)], whereas each measured distance is calculated after ~100 milliseconds of averaging. To clarify this point, we have made the following changes to the supplementary information:

Methods, page 9, line 182: For the smFRET measurements on the labelled constructs, the Cy5-lipid is confined to one side of the membrane for the duration of the measurement¹². The 0.002% of DOPE doped with DMPC lipids in our membrane composition has a diffusion coefficient of $\sim 0.56 \times 10^{-8} \text{ cm}^2\text{s}^{-1}$, which corresponds to a diffusion time of $\sim 1 \text{ ms}$ for the Cy5-lipid through the diameter of the disc¹³. Therefore, the extracted distances are a photon-weighted average over the rapid translational diffusion of the labeled dye across the surface of the membrane nanodisc.

Comment 2: What is the effect of any flexibility of the C-terminal tail on the observed FRET measurements. On what time scale does the c-terminal tail move? Does the width of the lifetime histograms reflect any flexibility?

Response 2: We thank the reviewer for this insightful question. The C-terminal tail rapidly rotates on a $\sim 100 \text{ ns}$ timescale [Lee, *Protein Sci.* (2006)], and so the measurement time for each point in the single-molecule FRET distributions ($\sim 100 \text{ ms}$) averages over this motion. We have clarified this point with the following addition to the supplementary information:

Methods, page 9, line 187: Previous anisotropic measurements on the intracellular domain fluorescently labeled at the C-terminus found a rotational time of ~85 ns,¹⁴ with reduced dynamics of the CTT upon EGF binding¹⁵. The measurement time for single-molecule FRET (~100-ms) averages over this flexible motion of the CTT.

Comment 3: Can you clarify whether the studies were conducted in the presence of ATP? Is the C-terminal tail phosphorylated in these experiments? And if it is, to what extent?

Response 3: We thank the reviewer for this question. The studies were not conducted in the presence of ATP. However, the cell-free extract used for the *in vitro* translation of full-length EGFR in nanodiscs contain both the ATP and metal ion cofactors necessary for phosphorylation of tyrosines in the EGFR C-terminal tail (Supplementary figure 2c).

Currently, the amount of phosphorylation cannot be quantified exactly in either of our two approaches to measure phosphorylation. In one approach, the detection of phosphorylated EGFR is based on measuring the fluorescence of the secondary antibody, so it also depends on the antibody binding efficiencies. In the other, the fluorescence of the anti-EGFR and anti-tyrosine blot cannot be compared to one another as the fluorophores used on the secondary antibodies for detecting EGFR and phosphorylated tyrosines are different in addition to the difference in the affinity of the primary and secondary antibodies to the binding site. We attempted to use mass spectrometry to perform such quantifications, but our protein levels are currently too low for clean signals. We hope such experiments will be possible in the future.

To clarify the presence of ATP in the cell-free extract, we made the following changes to the supplementary information:

Methods, page 2, line 16: Cell-free expression was carried out utilizing the Expressway Maxi Cell-Free *E.Coli* Expression system (Life Technologies), which contains both ATP and the metal ion cofactors necessary for tyrosine phosphorylation, to produce full-length EGFR fused with a SNAP tag (Supplementary Fig. 2a).

Comment 4: Related to (3), have you considered making mutations in the C-terminal tail (replace the tyrosine(s) with appropriately changes amino acids to mimic phosphorylation)

Response 4: We thank the reviewer for the suggestion. We can explore the effect of phosphorylation of the tyrosines in the EGFR C-terminal tail with the current platform. We agree such studies would be interesting, and hope to investigate these effects in the future.

As an initial investigation of this question, we performed smFRET experiments and simulations of an EGFR C-terminal construct with tyrosine 1173 chemically phosphorylated or dephosphorylated [Regmi, *J. Phys. Chem. Lett.* (2020)], and observed a conformational change upon phosphorylation. To refer to this study, we have added the above reference into the text as # 16 on page 3, line 36 of the updated manuscript.

Comment 5: Some of the earlier biophysical studies on the isolated intracellular domains seem to be missing from the reference list (the work by John Koland, for example)

Response 5: We thank the reviewer for this suggestion and the following references on the biophysical studies of isolated EGFR intracellular domain were added to the revised version of the manuscript.

1. Cheng K. & Koland, J.G. "Nucleotide Binding by the Epidermal Growth Factor Receptor Protein-tyrosine Kinase Trinitrophenyl-ATP as a spectroscopic probe". *Journal of Biological Chemistry*, 271(1), 311-318 (1996).
2. Lee, N. Y., & Koland, J. G. Conformational changes accompany phosphorylation of the epidermal growth factor receptor C-terminal domain. *Protein science*, 14, 2793--2803 (2005).
3. Lee N. Y., Hazlett T. L. & Koland J. G. "Structure and dynamics of the epidermal growth factor receptor C-terminal phosphorylation domain". *Protein science*, 15(5), 1142-1152 (2006).
4. Cadena D. L., Chan, C. L. & Gill, G. N. "The intracellular tyrosine kinase domain of the epidermal growth factor receptor undergoes a conformational change upon autophosphorylation". *Journal of Biological Chemistry*, 269, 260--265 (1994).
5. Landau, M., Fleishman, S. J. & Ben-Tal, N. A putative mechanism for downregulation of the catalytic activity of the EGF receptor via direct contact between its kinase and C-terminal domains. *Structure*, 12, 2265--2275 (2004).

The above are references 11-15 on page 3, line 36 of the updated manuscript.

Comment 6: From an allosteric view point do the data indicate an induced conformation produced by ligand or a selected conformation from a pre-existing ensemble of conformations

Response 6: We thank the reviewer for this comment. Our donor lifetime distributions of wildtype EGFR experiments appear bimodal, consistent with a model where the equilibrium between two states changes in the presence of the ligand. This indicates that the selection of a conformation is induced by a shift in a pre-existing equilibrium between at least two conformations.

We have clarified this point in the revised version of the manuscript.

Result, page 7, line 127: Our donor lifetime distributions of wildtype EGFR experiments appear bimodal, consistent with a model in which a pre-existing equilibrium between at least two conformations shifts in the presence of the ligand^{46,47}.

REVIEWERS' COMMENTS

Reviewer #2 (Remarks to the Author):

Author has addressed all the suggestions and concerns with utmost clarity. This manuscript can be accepted for publication.